# Stackelberg Self-Annotation:
# A Robust Approach to Data-Efficient LLM Alignment

**Xu Chu**[*] [1,2,3]**, Zhixin Zhang**[1,3]**, Tianyu Jia**[1,3]**, Yujie Jin**[1,3]

[1]Key Laboratory of High Confidence Software Technologies, Ministry of Education
[2]Center on Frontiers of Computing Studies, Peking University
[3]School of Computer Science, Peking University

 https://github.com/EunTilofy/SSAPO

## Abstract

Aligning large language models (LLMs) with human preferences typically demands vast amounts of meticulously curated data, which is both expensive and prone to labeling noise. We propose Stackelberg Game Preference Optimization (SGPO), a robust alignment framework that models alignment as a two-player Stackelberg game between a policy (leader) and a worst-case preference distribution (follower). The proposed SGPO guarantees $\mathcal{O}(\epsilon)$-bounded regret within an $\epsilon$-Wasserstein ball, offering formal robustness to (self-)annotation noise. We instantiate SGPO with Stackelberg Self-Annotated Preference Optimization (SSAPO), which uses minimal human-labeled "seed" preferences and iteratively self-annotates new prompts. In each iteration, SSAPO applies a distributionally robust reweighting of synthetic annotations, ensuring that noisy or biased self-labels do not derail training. Remarkably, using only 2K seed preferences—about 1/30 of standard human labels—SSAPO achieves strong win rates against GPT-4 across multiple benchmarks within three iterations. These results highlight that a principled Stackelberg formulation yields data-efficient alignment for LLMs, significantly reducing reliance on costly human annotations.

## 1 Introduction

Large language models (LLMs) have demonstrated remarkable capabilities across a broad range of tasks, but aligning their outputs with human preferences remains a core challenge for safety and usability [1, 2, 3]. Traditional alignment paradigms, such as Reinforcement Learning from Human Feedback (RLHF) [3, 4] or Direct Preference Optimization (DPO) [5], typically rely on large amounts of meticulously curated preference data. Such data collection is costly, time-consuming, and inevitably prone to labeling noise or bias, which can in turn degrade model performance once integrated at scale [6]. Consequently, an important question arises:*How can we achieve robust alignment of LLMs without relying on vast, error-prone human-labeled datasets?*

One promising direction is to reduce the need for human-annotated samples by having the model itself generate preference labels on newly sampled prompts—so-called "self-annotation" [7, 8, 9]. However, most self-annotation approaches overlook the fact that synthetic labels may be systematically biased or noisy. If these errors go unchecked, they can compound over iterative rounds of training, ultimately harming rather than helping alignment [10, 11].

In this work, we address this problem by **framing preference alignment as a two-player Stackelberg game** between a policy (leader) and a worst-case preference distribution (follower). Our formulation, which we call **Stackelberg Game Preference Optimization (SGPO)**, explicitly guards against plausible shifts or adversarial corruption in the preference data by operating within an $\epsilon$-Wasserstein ball [12] around the empirical distribution. We prove that the resulting policy achieves $\mathcal{O}(\epsilon)$-bounded

---

[*]Corresponding author. Contact E-mail: chu_xu@pku.edu.cn

39th Conference on Neural Information Processing Systems (NeurIPS 2025).

regret, a theoretical guarantee of robustness to noise or distribution mismatch [13]. This stands in contrast to standard DPO, whose regret can grow linearly with such shifts [5]. *We delay a more thorough **related work** section in the Appendix A.*

To instantiate SGPO with a concrete algorithm, we then present **Stackelberg Self-Annotated Preference Optimization (SSAPO)**. Starting from a small set of human-labeled "seed" preferences, SSAPO self-annotates new prompts by generating candidate responses and ranking them internally. Crucially, it couples this self-annotation with a distributionally robust reweighting [13] that prevents noisy synthetic labels from overwhelming the training updates. Remarkably, we find that using only 2K seed preference pairs (around 1/30 of the usual scale), SSAPO outperforms or matches methods that rely on significantly more human labels. On multiple alignment benchmarks—including AlpacaEval [14] and MT-Bench [15]—SSAPO rapidly achieves competitive or superior performance within just three rounds of iterative self-annotation.

We summarize our contributions as follows. 1. **Stackelberg formulation of preference alignment:** We recast alignment as a two-player game and prove the existence of a robust equilibrium with $\mathcal{O}(\epsilon)$-bounded regret under $\epsilon$-Wasserstein preference shifts. 2. **Robust self-annotation algorithm (SSAPO):** We instantiate our framework by combining minimal seed labels with iterative synthetic annotations. Our distributionally robust reweighting attenuates the impact of potential labeling noise. 3. **Data efficiency and empirical results:** Experiments show that SSAPO maintains high-level performance despite using only a fraction of typical human annotations, achieving strong results against GPT-4 in head-to-head comparisons.

## 2 Theoretical Foundation: SGPO Framework

This section formalizes *Stackelberg Game Preference Optimization* (SGPO) and establishes its guarantees. We begin with DPO preliminaries (Section 2.1), then cast SGPO as a two-player Stackelberg game over *gap distributions* (Section 2.2). We prove existence of a Stackelberg equilibrium and local convergence of a practical alternating scheme (Section 2.3), and finally quantify regret and contrast SGPO with DPO (Section 2.4). All proofs are deferred to Appendix D.

### 2.1 Preliminaries: Preference Datasets and DPO

**Preference-ranked dataset.** We use $D = \{(x^i, y_w^i, y_\ell^i)\}_{i=1}^N$, where $x^i$ is a prompt and $(y_w^i, y_\ell^i)$ are the *winner/loser* responses (from human or partially self-annotated feedback).

**RLHF and KL regularization.** Classical RLHF [4] optimizes a policy $\pi_\theta$ under a KL penalty to $\pi_{\mathrm{ref}}$:

$$\max_{\theta \in \Theta} \mathbb{E}_{x \sim \mathcal{D}, y \sim \pi_\theta(\cdot|x)} \left[ R(x,y) - \beta D_{\mathrm{KL}}\big(\pi_\theta(\cdot|x) \| \pi_{\mathrm{ref}}(\cdot|x)\big) \right], \tag{1}$$

where $\beta > 0$ controls regularization and $\mathcal{D}$ is the prompt distribution.

**Direct Preference Optimization (DPO).** Under the Bradley–Terry model $p(y_w \succ y_\ell \mid x) = \sigma(R(x, y_w) - R(x, y_\ell))$ with $\sigma(z) = 1/(1 + e^{-z})$, first-order optimality of a related KL-regularized objective yields

$$R(x,y) = \beta \log \frac{\pi_\theta(y|x)}{\pi_{\mathrm{ref}}(y|x)} + \beta \log Z(x), \tag{2}$$

with partition function $Z$. Plugging this into the BT likelihood gives

$$\mathcal{L}_{\mathrm{DPO}}(\theta) = \mathbb{E}_{(x, y_w, y_\ell) \sim D} \left[ \log \sigma\big( R(x, y_w) - R(x, y_\ell) \big) \right]. \tag{3}$$

DPO is simple but provides no explicit protection against shifts away from the empirical preference distribution. This motivates a robust formulation.

### 2.2 SGPO: A Two-Player Stackelberg Game

SGPO imposes robustness over *preference gaps*. For a policy $\pi$, define the *gap map*

$$\Delta R_\pi(x, y_w, y_\ell) := R_\pi(x, y_w) - R_\pi(x, y_\ell) \in \mathbb{R}.$$

Let $\hat{P} = \frac{1}{N}\sum_{i=1}^N \delta_{(x^i,y_w^i,y_\ell^i)}$ and let $(\Delta R_\pi)_\# \hat{P}$ be its push-forward, i.e.,

$$(\Delta R_\pi)_\# \hat{P} = \frac{1}{N}\sum_{i=1}^N \delta_{\Delta R_\pi(x^i,y_w^i,y_\ell^i)} \in \mathcal{P}(\mathbb{R}).$$

We measure uncertainty via the 1-Wasserstein ball $\mathcal{U}_\epsilon(\nu) := \{\alpha \in \mathcal{P}(\mathbb{R}) : W_1(\alpha,\nu) \le \epsilon\}$ centered at a gap distribution $\nu$. Hereafter, for simplicity, we write $\pi$ instead of $\pi_\theta$, the dependence on $\theta$ is implicitly assumed throughout the paper.

**Leader objective in gap space.** Let $f(\xi) = \log\sigma(\xi)$, a concave and 1-Lipschitz function on $\mathbb{R}$. The SGPO leader solves

$$\max_{\pi\in\Pi} \min_{\alpha\in\mathcal{U}_\epsilon\left((\Delta R_\pi)_\# \hat{P}\right)} \mathbb{E}_{\xi\sim\alpha}\big[f(\xi)\big]. \tag{4}$$

This objective optimizes the worst-case preference likelihood over perturbations in *gap space* rather than in token space. See Section 3 for implementation details.

**Follower best response in gap space.** For any center $\nu \in \mathcal{P}(\mathbb{R})$ and radius $\epsilon > 0$, define the follower best-response set

$$\mathcal{A}_\epsilon(\nu) := \arg\min_{\alpha\in\mathcal{P}(\mathbb{R})}\left\{\mathbb{E}_{\xi\sim\alpha}\big[f(\xi)\big] : W_1(\alpha,\nu) \le \epsilon\right\}. \tag{5}$$

When $\nu = (\Delta R_\pi)_\# \hat{P}$, any $\alpha^\star \in \mathcal{A}_\epsilon(\nu)$ is a follower best response against $\pi$.

This formalization induces the Stackelberg equilibrium:

**Definition 2.1** (Stackelberg equilibrium)**.** A pair $(\pi^*,\alpha^*)$ is a *Stackelberg equilibrium* if

$$\pi^* \in \arg\max_{\pi\in\Pi} \min_{\alpha\in\mathcal{U}_\epsilon\left((\Delta R_\pi)_\# \hat{P}\right)} \mathbb{E}_{\xi\sim\alpha}[f(\xi)], \qquad \alpha^* \in \mathcal{A}_\epsilon\big((\Delta R_{\pi^*})_\# \hat{P}\big). \tag{6}$$

This definition links the leader's robust optimization (4) with the follower's DRO problem (5). We next establish existence and analyze a practical alternating scheme.

### 2.3 Existence and Convergence of a Stackelberg Equilibrium

We first state mild conditions ensuring existence, then analyze an alternating best-response with a proximal leader step.

**Assumptions.** (i) $\Pi$ is compact. (ii) For each $(x,y)$, $\pi \mapsto R_\pi(x,y)$ is continuous; hence $\pi \mapsto (\Delta R_\pi)_\# \hat{P}$ is continuous in the weak topology. (iii) $f(\xi) = \log\sigma(\xi)$ is continuous and 1-Lipschitz on $\mathbb{R}$.

**Theorem 2.2** (Existence of a Stackelberg equilibrium)**.** *Under the assumptions above, problem* (4) *admits at least one solution* $(\pi^*,\alpha^*)$.

*Proof sketch.* For fixed $\pi$, the inner problem has a minimizer by compactness of $\mathcal{U}_\epsilon((\Delta R_\pi)_\# \hat{P})$ and continuity in $\alpha$. Berge's maximum theorem [16, 17] yields upper semicontinuity of $V(\pi) = \min_{\alpha\in\mathcal{U}_\epsilon((\Delta R_\pi)_\# \hat{P})} \mathbb{E}[f]$, and compactness of $\Pi$ gives a maximizer $\pi^*$ and a follower best response $\alpha^*$. See Appendix D. $\qquad\square$

**Alternating best responses with a proximal leader step.** Let $\hat{\alpha}(\pi) := (\Delta R_\pi)_\# \hat{P}$. Given $\pi_t$, choose a follower best response

$$\alpha_{t+1} \in \mathcal{A}_\epsilon\big(\hat{\alpha}(\pi_t)\big) \in \arg\min_{\alpha\in\mathcal{U}_\epsilon(\hat{\alpha}(\pi_t))} \mathbb{E}_{\xi\sim\alpha}[f(\xi)]. \tag{7}$$

Update the leader via a proximal step:

$$\pi_{t+1} \in \arg\max_{\pi\in\Pi}\left\{\min_{\alpha\in\mathcal{U}_\epsilon(\hat{\alpha}(\pi_t))} \mathbb{E}_{\xi\sim\alpha}[f(\xi)] - \frac{\lambda}{2}\|\pi - \pi_t\|^2\right\}. \tag{8}$$

This scheme makes the leader step stable while allowing the follower to track the changing center $\hat{\alpha}(\pi_t)$. The regularization term $\|\pi - \pi_t\|^2$ in the proximal step can be induced in practice with small learning rate and weight decay regularization.

**Theorem 2.3** (Well-posedness and local linear convergence). *Suppose the proximal leader objective is $\mu$-strongly concave in a neighborhood of $\pi^*$, uniformly over $\alpha \in \mathcal{U}_\epsilon(\hat{\alpha}(\pi))$, and $\pi \mapsto \hat{\alpha}(\pi)$ is locally Lipschitz in $W_1$. Then the update map $(\pi_t, \alpha_t) \mapsto (\pi_{t+1}, \alpha_{t+1})$ is a contraction near $(\pi^*, \alpha^*)$ and thus converges locally linearly.*

In practice, one may not directly implement (7)–(8), but the Theorem 2.3 shows that any procedure that approximates these alternating best-response updates can converge to the robust equilibrium. This provides a theoretical grounding for the SSAPO algorithm (to be introduced in the section 3), which combines standard gradient-based optimization with distributionally robust optimization.

## 2.4 Regret Analysis and Comparison with DPO

We quantify worst-case performance under gap-space shifts and compare with DPO. Define the performance functional
$$\mathcal{P}(\pi, \alpha) = \mathbb{E}_{\xi \sim \alpha}[\log \sigma(\xi)].$$
Let $\pi^\star$ solve (4). We prove that $\pi^*$ maintains high performance on *all* distributions $\alpha$ within $\epsilon$-Wasserstein distance of $\hat{\alpha}$. In particular, the drop from $\hat{P}$ to any $P$ is at most $\mathcal{O}(\epsilon)$.

**Theorem 2.4** (Worst-case performance in gap space). *For every $\alpha \in \mathcal{U}_\epsilon(\hat{\alpha}(\pi^\star))$,*

$$\mathcal{P}(\pi^\star, \alpha) \geq \mathcal{P}(\pi^\star, \hat{\alpha}(\pi^\star)) - \epsilon.$$

**Gap-space regret.** We define the Gap-space regret of a policy $\pi$ on a distribution $\alpha$ as $\text{Regret}(\pi, \alpha) := \max_{\tilde{\pi} \in \Pi} \mathcal{P}(\tilde{\pi}, \alpha) - \mathcal{P}(\pi, \alpha)$.

**Theorem 2.5** (SGPO regret bound). *For $\pi^\star$ solving (4),*

$$\sup_{\alpha \in \mathcal{U}_\epsilon(\hat{\alpha}(\pi^\star))} \text{Regret}(\pi^\star, \alpha) \leq 2\epsilon.$$

Thus, SGPO is robust: under any shift of at most $\epsilon$, its regret is bounded by a constant factor of $\epsilon$.

### 2.4.1 Comparison: DPO's Linear Regret

Let $\pi_{\text{DPO}} \in \text{argmax}_\pi \mathcal{P}(\pi, \hat{\alpha}(\pi))$ be a DPO solution on the empirical center. For any target $\alpha^\star$ with $\delta := W_1(\alpha^\star, \hat{\alpha}(\pi_{\text{DPO}}))$, we have:

**Theorem 2.6** (DPO regret lower bound). $\text{Regret}(\pi_{\text{DPO}}, \alpha^\star) \geq \delta - 2\epsilon$. *In particular, if $\delta \gg \epsilon$, DPO's regret grows linearly in $\delta$.*

**Corollary 2.7** (SGPO advantage over DPO). *If $W_1(\hat{\alpha}(\pi), \alpha^\star) = \delta > 2\epsilon$, then*

$$\frac{\text{Regret}(\pi_{\text{DPO}}, \alpha^\star)}{\text{Regret}(\pi^\star, \alpha^\star)} \geq \frac{\delta - 2\epsilon}{2\epsilon}.$$

Thus, SGPO's robust policy can outperform DPO by a factor of $\frac{\delta}{2\epsilon} - 1$ under sufficiently large distribution shift $\delta$. SGPO builds *in-sample* performance and *out-of-sample* robustness into a single objective by optimizing against $\mathcal{U}_\epsilon$ in gap space. The $\mathcal{O}(\epsilon)$ worst-case degradation contrasts with DPO's linear sensitivity to distribution mismatch $\delta$, mirroring our empirical results in Section 4.

## 3 Practical Instantiation: SSAPO Algorithm

We now present a practical and computationally tractable realization of the Stackelberg scheme from the theory section, called *Stackelberg Self-Annotated Preference Optimization (SSAPO)*. SSAPO implements the iterative leader–follower updates of Theorem 2.3 and (7)–(8) for preference alignment.

**Notation.** Let $\sigma(u) = (1 + e^{-u})^{-1}$ and define the *margin* random variable

$$\xi := \Delta R_\theta(x, y_w, y_\ell) \quad \text{for winner–loser pairs } (y_w, y_\ell), \quad \hat{\alpha}(\pi_t) = \frac{1}{N} \sum_{i=1}^{N} \delta_{\hat{\xi}_i}, \ \hat{\xi}_i := \Delta R_{\theta_t}(x^i, y_w^i, y_\ell^i).$$

The follower (adversary) chooses a distribution $\alpha$ over $\xi$ inside the $W_1$ Wasserstein ball $\mathcal{U}_\epsilon(\hat{\alpha}(\pi_t))$ centered at $\hat{\alpha}(\pi_t)$, while the leader (policy) updates $\theta$.

**Implementation challenges addressed.** 1. *Minimal human labels via self-annotation.* We bootstrap from a small seed of human-labeled preferences and enlarge the dataset by letting the current policy rank its own responses on unlabeled prompts. 2. *Loss re-representation for tractable DRO.* The follower minimizes a concave inner objective $\mathbb{E}_\alpha[\log\sigma(\xi)]$. Writing $\ell(\xi) := -\log\sigma(\xi)$, which is convex and 1-Lipschitz, turns the inner problem into $-\sup_\alpha \mathbb{E}_\alpha[\ell(\xi)]$ and allows a *convex* PWL surrogate that yields a finite convex program for the follower. 3. *Scalability via uniform grouping.* For large datasets, we solve the follower subproblem on groups and average the resulting worst-case distributions, trading a small approximation for substantial speed-ups.

## 3.1 Follower objective: loss re-representation and a closed form

Since $\log\sigma$ is concave and 1-Lipschitz, define the convex 1-Lipschitz loss

$$\ell(\xi) := -\log\sigma(\xi) = \log(1+e^{-\xi}), \qquad \min_{\alpha\in\mathcal{U}_\epsilon(\hat{\alpha}(\pi_t))} \mathbb{E}_\alpha[\log\sigma(\xi)] = - \sup_{\alpha\in\mathcal{U}_\epsilon(\hat{\alpha}(\pi_t))} \mathbb{E}_\alpha[\ell(\xi)].$$

When $\Xi = \mathbb{R}$ (no support restrictions) and the ground metric is the absolute value, the worst-case expectation of any $L$-Lipschitz function equals the empirical mean plus $L\epsilon$. Specializing this *convex reduction* result of Mohajerin Esfahani and Kuhn [13, Thm. 6.3] to $\ell$ (whose Lipschitz constant is 1) gives:

**Lemma 3.1** (Closed-form follower in the unconstrained one-dimensional case; 13, Thm. 6.3)**.** *If* $\Xi = \mathbb{R}$ *and* $\mathcal{U}_\epsilon$ *is a* $W_1$ *ball (absolute ground metric), then*

$$\sup_{\alpha\in\mathcal{U}_\epsilon(\hat{\alpha}(\pi_t))} \mathbb{E}_\alpha[\ell(\xi)] = \tfrac{1}{N}\sum_{i=1}^{N}\ell(\hat{\xi}_i) + \epsilon, \qquad \Longleftrightarrow \qquad \min_\alpha \mathbb{E}_\alpha[\log\sigma(\xi)] = \tfrac{1}{N}\sum_{i=1}^{N}\log\sigma(\hat{\xi}_i) - \epsilon.$$

Although Lemma 3.1 provides an exact closed form in the unconstrained 1-D setting, in this paper we propose to solve a *finite convex program* that also returns a discrete worst-case distribution.

## 3.2 Follower via a max-of-affine surrogate and a finite convex program

We approximate $\ell$ by a convex piecewise-linear under-approximation

$$\tilde{\ell}(\xi) = \max_{1\leq k\leq K}\{a_k\xi+b_k\} \leq \ell(\xi), \quad \text{with} \quad a_k = \ell'(\xi^{(k)}) = -\sigma(-\xi^{(k)}), \; b_k = \ell(\xi^{(k)}) - a_k\xi^{(k)}.$$

Knots $\{\xi^{(k)}\}_{k=1}^K$ are chosen on a window $[a_t, b_t]$ in margin space (empirical quantiles or $[\min_i\hat{\xi}_i - \tau, \max_i\hat{\xi}_i + \tau]$ with small $\tau > 0$). Endpoint tangents extend $\tilde{\ell}$ outside $[a_t, b_t]$, preserving $\tilde{\ell}\leq\ell$ globally. Because $\tilde{\ell}\leq\ell$, replacing $\ell$ by $\tilde{\ell}$ in (3.1) yields an *upper bound* on the original inner minimum, which tightens with $K$:

**Proposition 3.2** (Monotone tightening in $K$)**.** *Let $\hat{\alpha}$ be a probability measure on $\mathbb{R}$ with finite first moment and let $\mathcal{U}_\epsilon(\hat{\alpha})$ be the $W_1$ ball of radius $\epsilon\geq 0$. Let $\ell(\xi) = -\log\sigma(\xi)$ and let $(\tilde{\ell}_K)_{K\geq 1}$ be convex piecewise-linear underestimators of $\ell$ such that $\tilde{\ell}_K \leq \tilde{\ell}_{K+1}\leq\ell$ pointwise and $\tilde{\ell}_K\uparrow\ell$. Define*

$$v^\star := \inf_{\alpha\in\mathcal{U}_\epsilon(\hat{\alpha})} \mathbb{E}_\alpha[\log\sigma(\xi)] = - \sup_{\alpha\in\mathcal{U}_\epsilon(\hat{\alpha})} \mathbb{E}_\alpha[\ell(\xi)], \quad v_K := - \sup_{\alpha\in\mathcal{U}_\epsilon(\hat{\alpha})} \mathbb{E}_\alpha[\tilde{\ell}_K(\xi)].$$

*Then, for all $K\geq 1$: (i) $v_K \geq v^\star$ (valid upper bound); (ii) $v_{K+1}\leq v_K$ (monotone in $K$); (iii) $v_K\downarrow v^\star$ as $K\to\infty$.*

**A finite convex program for the worst case (after 13, Thm. 4.4).** For losses representable as a *pointwise maximum of finitely many affine functions*,

$$\sup_{\alpha\in\mathcal{U}_\epsilon(\hat{\alpha}(\pi_t))} \mathbb{E}_\alpha\Big[\max_{k\leq K}\{a_k\xi+b_k\}\Big]$$

admits a *finite convex program* whose solution is a *discrete extremal distribution*. Specializing Mohajerin Esfahani and Kuhn [13, Thm. 4.4] to our one-dimensional $\xi$ and absolute ground metric yields:

**Theorem 3.3** (Finite convex program for max-of-affine (PWL convex) losses; specialization of Mohajerin Esfahani and Kuhn [13], Thm. 4.4)**.** *Let $\tilde{\ell}(\xi) = \max_{k\leq K}\{a_k\xi+b_k\}$ and $\Xi\subseteq\mathbb{R}$. Introduce, for each sample $i$ and piece $k$, a mixing weight $s_{ik}\geq 0$ and a displacement $q_{ik}\in\mathbb{R}$. Then*

$$\sup_{\alpha\in\mathcal{U}_\epsilon(\hat{\alpha}(\pi_t))} \mathbb{E}_\alpha[\tilde{\ell}(\xi)] = \max_{\{s_{ik},q_{ik}\}} \frac{1}{N}\sum_{i=1}^{N}\sum_{k=1}^{K}\Big[s_{ik}(a_k\hat{\xi}_i+b_k) - a_kq_{ik}\Big] \qquad (9)$$

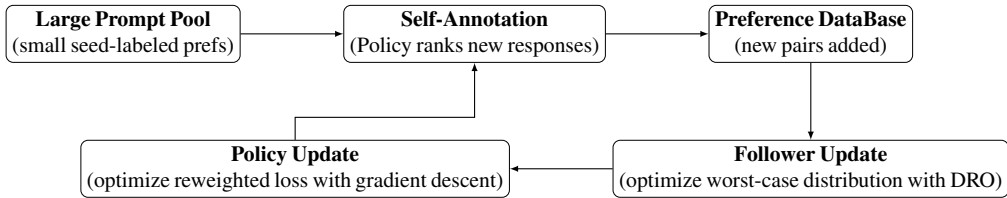

Figure 1: **SSAPO workflow.** We maintain a large prompt pool and a small set of seed-labeled preferences. The policy self-annotates new prompts by generating and ranking responses, thus expanding the preference database. A follower then identifies a worst-case distribution for these preferences, and the leader (policy) is updated accordingly. This process repeats for iterations.

*subject to the Wasserstein and feasibility constraints*

$$\frac{1}{N}\sum_{i=1}^{N}\sum_{k=1}^{K}|q_{ik}| \leq \epsilon, \qquad \sum_{k=1}^{K}s_{ik}=1, \ s_{ik}\geq 0 \quad (\forall i), \qquad a_t s_{ik} \leq s_{ik}\hat{\xi}_i - q_{ik} \leq b_t s_{ik} \quad (\forall i,k).$$

*An extremal discrete measure* $\alpha_t^* = \frac{1}{N}\sum_{i=1}^{N}\sum_{k=1}^{K}s_{ik}^*\delta_{\hat{\xi}_i - q_{ik}^*/s_{ik}^*}$ *attains the supremum.*

The change of variables $z_{ik} = \hat{\xi}_i - q_{ik}/s_{ik}$ (when $s_{ik} > 0$) reveals that $s_{ik}$ splits the unit mass at $\hat{\xi}_i$ across pieces, while $q_{ik}$ transports that mass in margin space. The interval constraints are linear "perspective" constraints enforcing $\xi \in [a_t, b_t]$.

### 3.3   SSAPO workflow

Figure 1 summarizes the SSAPO workflow. Starting with a small seed of human-labeled preferences plus a large unlabeled pool, we proceed in the following loop, at iteration $t$:

1. **Self-annotation.** Sample prompts from the unlabeled pool, generate multiple responses, and let the current policy $\pi_{\theta_t}$ rank them to create new preference pairs $(y_w, y_\ell)$.
2. **Empirical center.** Compute $\hat{\xi}_i = \Delta R_{\theta_t}(x^i, y_w^i, y_\ell^i)$ and set $\hat{\alpha}(\pi_t) = \frac{1}{N}\sum_i \delta_{\hat{\xi}_i}$.
3. **Follower (DRO).** Build $\tilde{\ell}(\xi) = \max_{k \leq K}(a_k \xi + b_k)$ using tangents at knots $\{\xi^{(k)}\}$; choose $[a_t, b_t]$ as above. Solve the convex program in Theorem 3.3 to obtain the worst-case $\alpha_t^*$.
4. **Leader update.** Update $\theta$ by minimizing
$$\theta_{t+1} \in \operatorname*{argmin}_{\theta} \mathbb{E}_{(x,y_w,y_\ell)\sim\mathcal{D}}\mathbb{E}_{\xi\sim\alpha_t^*}\left[\ell\big(\Delta R_\theta(x,y_w,y_\ell)\big)\right],$$
   implemented as standard minibatch SGD on the preference pairs while incorporating the per-sample mixture weights induced by $(s^*, q^*)$.

Repeating for $T$ total iterations yields the final aligned policy $\pi_{\theta_T}$. A more explicit version of SSAPO is provided in Algorithm 1 (Appendix E), along with its computational complexity analysis.

**How $\alpha_t^*$ enters the leader step.**   The optimizer $(s^*, q^*)$ associates to each training pair $i$ a set of active affine pieces (weights $\{s_{ik}^*\}_k$) and a transport direction (through $q_{ik}^*/s_{ik}^*$). In practice we (i) weight the per-pair loss contributions by $\{s_{ik}^*\}_k$ and (ii) optionally add a proximal penalty nudging the current margin toward $\hat{\xi}_i - q_{ik}^*/s_{ik}^*$ for stability.

### 3.4   Scalability and Complexity

**Grouping for large $N$.**   When $N$ is large (e.g. $10^5$ or more preferences), solving the convex program in Step (Worst-Case Distribution) can be expensive. A popular heuristic partitions $\{\hat{\xi}_1, ..., \hat{\xi}_N\}$ into $M$ groups (each of size $G = N/M$), and solves the finite program (9) separately within each group.

The resulting distributions $\alpha_m^*$ are then averaged (or merged proportionally):

$$P_{\text{final}}^* = \frac{1}{M}\sum_{m=1}^{M}\alpha_m^*.$$

While not an *exact* solution to the global $N$-sample problem, this still confers substantial robustness while reducing complexity from $N$ to $G \ll N$ in each subproblem. In summary, this grouping approach greatly reduces memory/compute cost, and is parallelizable. Section F in the appendix remarks the approximation effects of SSAPO algorithm design on SGPO guarantees

**Complexity.** The program in Theorem 3.3 introduces $O(NK)$ variables and $O(NK)$ constraints and becomes a pure LP after linearizing $|q_{ik}|$ via standard slack variables. With warm starts, per-iteration time scales nearly linearly in $N$ and linearly in $K$. A more detailed algorithmic complexity analysis is in Appendix E.

**Why a finite convex program is necessary in practice.** We argue the special case of unconstrained one–dimensional case the follower (that collapses to an exact closed form from Lemma 3.1) is too restrictive for real pipelines: we (i) clip or restrict the support of margins for stability, (ii) require extremal *discrete* adversaries whose mixture weights can be recycled to form stochastic gradients for the leader, and (iii) need a controllable approximation whose accuracy improves monotonically. Accordingly, SSAPO replaces the inner concave expectation $\min_{\alpha \in \mathcal{U}_\epsilon(\hat{\alpha}(\theta_t))} \mathbb{E}_\alpha[\log \sigma(\xi)]$ by a tractable *finite convex program* obtained from a convex piecewise-linear surrogate $\tilde{\ell}_K(\xi) = \max_{k \leq K} \{a_k \xi + b_k\}$ of $\ell(\xi) = -\log \sigma(\xi)$. This yields an upper bound $-\sup_\alpha \mathbb{E}_\alpha[\tilde{\ell}_K(\xi)]$ on the inner minimum that tightens monotonically in $K$ (Proposition 3.2) and becomes exact as $K \to \infty$. Moreover, specializing Mohajerin Esfahani and Kuhn [13, Thm. 4.4] to our one–dimensional absolute ground metric produces a finite convex program whose optimizer is a *discrete* extremal distribution supported on at most $NK$ atoms: exactly the structure we need to implement sample reweighting and efficient leader updates. In short, the finite convex program is both *theoretically correct* (via 13, Thm. 6.3 and Thm. 4.4) and *operationally necessary* for SSAPO's **stability** and efficiency.

# 4 Experiments

In this section, we present an extensive empirical evaluation of our proposed *Stackelberg Self-Annotated Preference Optimization* (SSAPO) algorithm.

## 4.1 Experiment Setup

We introduce the basic experiment setup in this subsection (Cf. Appendix G for more details). The settings are mostly consistent to the recent literature Kim et al. [9]. **Datasets**. We used the UltraFeedback dataset [18], containing 60K samples. A seed of 2K human-labeled preferences (3.3% of total 60K data) was used for initial training. The rest (58K samples) were split into three subsets (8K, 20K, and 30K) for self-annotation in iterative stages.

**Models**. We use the supervised fine-tuned Mistral-7B-0.1 [19] as the initial model $\pi_{\text{init}}$ and LLaMA-3-8B[2] for compatibility checks. All models are fine-tuned on UltraChat [20].

**Evaluations**. We use **AlpacaEval 2.0** [14] for instruction-following tasks and **MT-Bench** [15] to evaluate multi-turn performance across tasks like math, coding, and writing. Both benchmarks assess the alignment with human preferences and the model's functional proficiency. We stress that AlpacaEval 2.0 is especially useful for measuring how well the model aligns with general user preferences (and controlling for length bias), whereas MT-Bench tests the model's functional capabilities across a broader range of tasks.

**Implementation.** We initialize training with DPO on 2K seed samples, followed by 3 iterative stages of self-annotation. In each stage, new preferences are generated via a policy that ranks response pairs. A distributionally robust optimization (DRO) is performed using sequential least squares programming (SLSQP) to adjust the model based on adversarial shifts within a Wasserstein ball. The group size $G$ for parallel computation is set to 100 unless otherwise specified.

**Baselines.** We consider the following baselines for comparison: (1) DPO, which performs DPO training only on the seed data. (2) Iter DPO [11], which iteratively generates preference data using an external reward model (PairRM) [21] or LLM-as-judge [22]. (3) SPA [9], which iteratively generates preference data using implicit reward model.

[2]meta-LLaMA/Meta-LLaMA-3-8B-Instruct

Table 1: **Main results.** Evaluation results on AlpacaEval 2.0 and MT-Bench with different variants of Mistral-7B-v0.1 and LLaMA3-8B. All models use the same 2K preference data with gold label as seed data. The best and second-best results are highlighted in bold and underlined, respectively. Most of the baseline results are from [9].

| Models | AlpacaEval 2.0 | | MT-Bench |
|---|---|---|---|
| | Len-control. Win Rate (%) | Win Rate vs. GPT-4 (%) | Avg. Score (0-10) |
| Mistral-7B-DPO | 9.03 | 7.68 | 6.81 |
| Mistral-7B-Iter DPO (PairRM) | 11.87 | 9.46 | 6.98 |
| Mistral-7B-Iter DPO (LLM-as-judge) | 9.28 | 9.18 | 6.67 |
| LLaMA3-8B-DPO | 20.61 | 18.04 | - |
| Mistral-7B-SPA | 15.39 | 21.13 | 6.94 |
| LLaMA3-8B-SPA | 21.85 | 24.95 | 7.86 |
| Mistral-7B-SSAPO (Ours) | 24.44 | 35.82 | 6.68 |
| LLaMA3-8B-SSAPO (Ours) | **33.33** | **40.12** | **8.03** |

Table 2: **Comparison with different variants of Mistral.** Evaluation results on AlpacaEval 2.0 and MT-Bench with different variants of Mistral-7B-v0.1. The best scores are highlighted with bold. The baseline results are from [9] and [23].

| Models | Gold Label (%) | AlpacaEval 2.0 | | MT-Bench |
|---|---|---|---|---|
| | | Len-control. Win Rate (%) | Win Rate vs. GPT-4 (%) | Avg. Score (0-10) |
| Mistral-7B-v0.1 | - | 0.17 | 0.50 | 3.25 |
| Zephyr-7B-$\beta$ | 100 | 11.75 | 10.03 | 6.87 |
| Mistral-7B-SFT | - | 7.58 | 4.72 | 6.34 |
| Mistral-7B-DPO | 3.3 | 9.03 | 7.68 | 6.81 |
| Mistral-Large (123B) | - | 21.4 | 32.7 | - |
| Mistral-7B-SSAPO (Ours) | 3.3 | **24.44** | **35.82** | 6.68 |

## 4.2 Main Results

Table 1 summarizes our primary comparison on **AlpacaEval 2.0** and **MT-Bench**. All models in this comparison use only 2K preference pairs of the UltraFeedback dataset as seed data (3.3% out of 60K), with the remainder self-annotated. Our *SSAPO* method consistently outperforms DPO and other iterative baselines (e.g., Iter-DPO, SPA) in both the length-controlled (LC) and raw win-rate metrics on AlpacaEval 2.0. For Mistral-7B, **SSAPO** achieves 24.44% LC win rate and 35.82% raw win rate, compared to only 9.03% and 7.68% with standard DPO. On the larger LLaMA-3-8B model, SSAPO reaches a **33.33%** LC win rate and **40.12%** raw win rate, surpassing its SPA counterpart by a wide margin. MT-Bench scores corroborate these improvements, indicating that SSAPO yields robust, high-quality responses across diverse tasks.

To further illustrate SSAPO's data-efficiency and robustness, Table 2 compares various Mistral models, including *Mistral-7B-SFT*, *Mistral-Large* (the number of parameters is 123B), and a fully-finetuned *Zephyr-7B-$\beta$* variant with 100% labeled data. Remarkably, **Mistral-7B-SSAPO** outperforms or closely approaches these stronger references in AlpacaEval 2.0, despite using only 2K preference pairs (3.3% out of the 60K human-labeled training set). This demonstrates that *a principled Stackelberg method can substantially mitigate the reliance on massive human annotations*. It also aligns with our theoretical findings (Section 2) that SGPO-based approaches, when instantiated as SSAPO, achieve bounded regret under moderate preference shift.

## 4.3 Ablation and Sensitivity Analysis

Table 3: **Effect of Wasserstein Radius $\epsilon$ on Performance.** Evaluation results on Mistral-7B, showing the impact of varying the Wasserstein radius on the Len-control. Win Rate and Win Rate vs. GPT-4.

| $\epsilon$ | 0 | 0.01 | 0.03 | 0.05 | 0.1 |
|---|---|---|---|---|---|
| Len-control. Win Rate (%) | 19.76 | **24.44** | 22.42 | 23.20 | 19.78 |
| Win Rate vs. GPT-4 (%) | 26.58 | **35.82** | 32.30 | 32.92 | 25.84 |

We conduct a series of ablation studies to understand the factors influencing the efficacy and robustness of our *Stackelberg Self-Annotated Preference Optimization* (SSAPO). Specifically, we vary the

Table 4: **Impact of Tangent Size ($K$) and Impact of Group Size ($G$) on Model Performance.**
Evaluation results on Mistral-7B.

| Impact of Tangent Size ($K$) | | | | Effect of Group Size ($G$) | | | |
|---|---|---|---|---|---|---|---|
| $K$ | 5 | 6 | 7 | $G$ | 100 | 200 | 300 |
| **Len-control. Win Rate (%)** | 22.89 | **23.20** | 19.05 | **CPU Runtime (min)** | 45 | 206 | 630 |
| **Win Rate vs. GPT-4 (%)** | 29.19 | **32.92** | 25.84 | **Len-control. Win Rate (%)** | 13.70 | 14.81 | **16.95** |
| | | | | **Win Rate vs. GPT-4 (%)** | 10.00 | 11.74 | **14.91** |

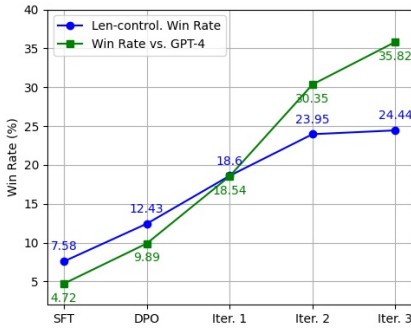

Figure 2: **Improvement during iterations** Evaluation results on AlpcaEval 2.0 of initial DPO stage and each iteration, the results of the SFT model are from [9].

Wasserstein radius $\epsilon$, the number of tangents $K$, and the group size $G$. We conduct the experiments on the Mistral-7B model for budget consideration. These experiments confirm our method's flexibility and validate the practical design choices guided by our theoretical framework.

**Wasserstein Radius $\epsilon$.** Table 3 demonstrates how performance changes with different Wasserstein radius. When $\epsilon = 0$, our approach reduces to self-annotated DPO without robust reweighting, yielding weaker results (19.76% LC win rate). As $\epsilon$ increases slightly (e.g., 0.01–0.05), both win-rates improve substantially, with the best outcomes at $\epsilon = 0.01$. However, overly large $\epsilon$ (e.g., 0.1) can make the adversarial shift too pessimistic, degrading performance. These findings align with our theoretical analysis in Section 2, where moderate $\epsilon$ provides a robust yet not overly conservative solution, thus striking the optimal balance between data fidelity and adversarial resilience.

**Number of Tangents $K$.** Since our piecewise-linear approximation of $-\log \sigma(\cdot)$ uses $K$ linear segments (cf. Section 3), we examine how varying $K$ affects alignment (Table 4 left). At $K = 5$, the model attains a 22.89% LC win-rate, while increasing to $K = 6$ yields a marginally better 23.20%. Interestingly, moving to $K = 7$ leads to performance drops (19.05%). We hypothesize that while a larger $K$ refines the convex under-approximation, it may also overcomplicate optimization or amplify minor errors in the approximation. Thus, $K = 6$ serves as a sweet spot in our setting, balancing expressiveness and computational stability.

**Group Size $G$.** Our distributionally robust optimization solver randomly partition data into groups of size $G$ for parallel subproblem solutions. Table 4 (right half) illustrates the trade-off between computational cost and performance. A small group size ($G = 100$) has faster runtime (45 min) but yields a 13.70% LC win-rate, whereas a larger $G = 300$ reaches 16.95% yet takes over 10 times longer (630 min). This confirms that while bigger groups permit more fine-grained reweighting and hence improved alignment, the overhead grows significantly. In practice, we choose $G = 100$ or $G = 200$ for an acceptable performance–efficiency balance.

**Iterative Performance Gains.** Figure 2 provides a direct illustration of iterative improvement over three rounds of SSAPO. Starting from a baseline DPO model, each round not only adds new self-annotated preferences but also reweights them adversarially within an $\epsilon$-ball. We observe a consistent upward trend in alignment metrics during the first two rounds, validating our claim that *robust self-annotation can compensate for scarce human labels while preserving alignment quality.*

Taken together, these ablations highlight the flexibility and effectiveness of SSAPO: Moderate $\epsilon$ balances robustness and data fidelity, confirming our theoretical finding that worst-case reweighting within a bounded radius can significantly enhance alignment without over-penalizing feasible distributions.Piecewise-linear approximations with small $K$ are sufficient to capture the shape of $-\log(\sigma(\cdot))$, maintaining computational tractability. Group size $G$ offers a controllable trade-off between runtime and fine-grained adversarial reweighting, making the approach scalable to different budget constraints. Iterative self-annotation with minimal seed data substantially boosts alignment, demonstrating that only 2K human-labeled preferences can suffice to achieve high performance. Overall, these experiments affirm our primary contributions: a *data-efficient* and *theoretically grounded* approach to preference alignment.

> **Practical Hyperparameter Guidelines**
>
> **Wasserstein radius $\epsilon$.** Scale $\epsilon$ with expected self-annotation noise. **Capable models:** $\epsilon \in [0.005, 0.02]$; **smaller models:** $\epsilon \in [0.01, 0.05]$. A robust default is $\epsilon = 0.01$ when no validation is available.
> **Piecewise approximation tangents $K$.** We recommend $K = 6$ as a stable default. $K = 7$ may *hurt* due to solver instability rather than approximation error.
> **Grouping size $G$.** For parallel DRO, $G \in [100, 1000]$ balances robustness and throughput; we find $G \approx 100$–$300$ a sweet spot in practice.

### 4.3.1 Robustness to Seed Label Noise (25% flips)

To assess robustness promised by our $O(\epsilon)$-regret guarantee, we flip the preferred/unpreferred labels on **25%** of the **2K** seed pairs and re-run SSAPO end-to-end. Table 5 shows that **Mistral-SSAPO** suffers only a $\sim$7–13% degradation, while **LLaMA-SSAPO** improves under noise, indicating DRO regularization and stronger self-annotation can counteract moderate seed noise.

Table 5: **Effect of 25% seed label corruption.** Entries show (AlpacaEval 2.0 LC win-rate / Win-rate vs GPT-4), higher is better.

| Model | No noise | 25% noise |
|---|---|---|
| Mistral-SSAPO | 26.90% / 31.93% | 19.70% / 18.51% |
| LLaMA-SSAPO | 33.33% / 40.12% | 43.74% / 46.70% |

*Discussion.* For Mistral, bounded degradation empirically aligns with our $O(\epsilon)$-regret theory. For LLaMA-3-8B, noise acts as implicit regularization: the worst-case distribution explores a wider $\epsilon$-ball region and mitigates overfitting to small seeds.

## 5  Conclusion, Limitations and Future Work

Aiming at a data-efficient alignment method, we have introduced SGPO alignment framework with $\mathcal{O}(\epsilon)$-bounded regret under moderate noise or distribution shifts. Our practical instantiation, *SSAPO*, uses self-annotation and distributionally robust reweighting to achieve strong performance with far fewer human labels. The scalability limitation of SSAPO comes from the number of preferences $N$, we use a simple uniform group trick to balance between robustness and complexity. For further improvement, one may resort to primal-dual or specialized cutting-plane methods [13], or use approximate relaxations with entropic regularization [24]. Our guarantees target *training-time* robustness to *preference noise* and mild distribution shifts (e.g., reweightings within a $W_1$ $\epsilon$-ball). This is distinct from *inference-time* robustness to adversarial prompts or jailbreak attacks. While our DRO step improves alignment under noisy supervision, it does not replace dedicated safety mechanisms for adversarial inputs. We only consider the human-labeled preference restricted scenario, however, SSAPO can also be integrated with prompt-generation procedure such as EVA [25], which could be crucial to scaling large language model based intelligence, considering that high-quality human data is running out in the next few years [26].

## Acknowledgments

We thank the reviewers for their useful feedback that improved this work. This research is supported by the National Natural Science Foundation of China (NSFC) under grants No.62506010.

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

**Organization of the Appendix.**

- **Section A** recap LLM alignment and data-efficient methods, as well as the Game-theoretic alignment methods. And dicuss the connection and distinction between SGPO/SSAPO with them.

- **Section B** revisits the core definitions and properties of the 1-Wasserstein metric, including a statement of the Kantorovich–Rubinstein duality.

- **Section C** restates and discusses the regularity conditions needed for our theoretical guarantees, such as compactness and Lipschitz continuity.

- **Section D** provides Proofs for the existence and convergence of the Stackelberg equilibrium, as well as the regret bounds for SGPO and comparisons with DPO.

- **Section E** presents the SSAPO algorithm in pseudocode and includes an analysis of its computational complexity.

- **Section F** remarks the approximation effects of SSAPO algorithm design on SGPO guarantees

- **Section G** gives supplementary information on experimental setups, hyperparameter choices, grouping strategies for DRO, and other implementation details.

- **Section H** illustrates additional qualitative comparisons of model outputs, highlighting the differences between DPO, SPA, and SSAPO in practice.

- **Section I** discusses the potential broader impact of this work.

## A    More Detailed Related Work

**LLM Alignment and Data-Efficient Methods**    Aligning large language models (LLMs) with human preferences is central to modern deployments [1, 2, 3],. While Reinforcement Learning with Human Feedback (RLHF) [4] trains a reward model and then maximizes it under KL constraints, it typically requires massive human-annotated data. Recent alternatives focus on *directly* fine-tuning LLMs from pairwise preference data without an explicit reward model. Notably, Direct Preference Optimization (DPO) [5] derives a closed-form surrogate objective that recovers RLHF's solution but avoids a separate reward modeling stage. Subsequent works simplify or extend this pipeline; for instance, Ethayarajh et al. [27] remove the need for pairwise labels by adopting a human utility model, while there are also works [28, 29, 30] introduce novel optimization objectives to handle different preference formats. Despite progress, these approaches still rely on large-scale preference annotations, making label-efficiency a key challenge. To reduce the reliance on expensive human labels, several methods have explored letting the LLM or an auxiliary model generate and rank unlabeled responses, thereby creating synthetic preference data [21, 8, 11, 9]. However, many of these approaches assume accessibility to a reliable well-aligned "judge", which could be prohibitive costly in realistic scenarios. To address the cost bottleneck, Kim et al. [9] propose a *Spread Preference Annotation (SPA)* framework that starts from a small seed of human-annotated preferences and iteratively expands the dataset by self-annotation. Our work is closely related to SPA: we replicate its experimental setup by using the same small-scale seed preferences and iterating between new response generation and preference learning. However, our *Stackelberg* perspective considers the inaccuracy of self-annotation, and explicitly defends against worst-case preference shifts. Empirically, we show that this game-theoretic *distributional* approach yields stronger label efficiency.

**Game-Theoretic Alignment Methods**    An emerging body of work has begun to frame preference alignment of LLMs through the lens of *games*. A conceptual similar work [31] propose *Stackelberg Alignment RLHF*. However, their nested gradient-based heuristic does not guaranteed to converge to the equilibrium. While we prove our updates for the leader and follower converge to an equilibrium. Meanwhile, Ye et al. [25] present a framework that casts prompt-creator and solver asymmetric players in an evolving game, the differences between our work is we focus on evolving the distribution of the responses , while they focus on evoling the distribution of the prompts. SPIN [32] use self-play to iteratively refine a policy without additional human data, however they assume accessible to adequate supervised fine-tuning (SFT) data. Other works adopt *Nash* or *minimax* formulations: Melnyk et al. [33] study alignment via an optimal-transport objective to capture distributional preferences, Zhang et al. [34] and Rosset et al. [35] formulate alignment as a two-player game aiming for a Nash policy, and Munos et al. [36] proposes "Nash learning from human feedback" by treating the policy and a competing policy as iterative players. Likewise, Swamy et al. [37], Wu et al. [38] introduce self-play preference

optimization methods in which two policies repeatedly compete under a constant-sum setting. They demonstrate promising performance on synthetic and text-based benchmarks, but typically set both players as *policy vs. policy*. By contrast, our *SGPO* framework focuses on *policy vs. distribution*: the leader policy maximizes preference likelihood, while the follower adversarially reweights or shifts the empirical preference distribution. This setup offers a distinct distributional robust-control view, leading to tight theoretical guarantees (e.g., $\mathcal{O}(\epsilon)$-bounded regret) and a practical algorithm (SSAPO) that is readily integrated with self-annotation. Hence, our method complements the "policy vs. policy" family by delivering strong resistance to noisy or distribution-mismatched preferences at small annotation cost.

# B    Preliminaries on the Wasserstein Metric Space

Wasserstein (or Earth Mover's) distances are widely used in robust optimization and optimal transport to measure how far two probability distributions are from one another [12]. Below, we give a concise overview of the 1-Wasserstein metric on a subset $\Xi \subseteq \mathbb{R}^m$. We also recap the Kantorovich–Rubinstein duality (Lemma B.2), which is central to several of our regret and robustness proofs in the main text.

## B.1    Definition of the 1-Wasserstein Metric

Let $\mathcal{M}(\Xi)$ be the space of all probability distributions supported on $\Xi$ such that

$$\mathbb{E}_{\xi \sim F}\big[\|\xi\|\big] = \int_{\Xi} \|\xi\| \, dF(\xi) < \infty.$$

In our setting, $\|\cdot\|$ can be any norm on $\mathbb{R}^m$, typically the Euclidean norm (although other choices are possible).

**Definition B.1** (1-Wasserstein Metric)**.** For two probability distributions $F_1, F_2 \in \mathcal{M}(\Xi)$, the *1-Wasserstein* distance (often just called "the Wasserstein distance") is defined as

$$W_1(F_1, F_2) := \inf_{\pi \in \Pi(F_1, F_2)} \left\{ \int_{\Xi \times \Xi} \|\xi_1 - \xi_2\| \, d\pi(\xi_1, \xi_2) \right\}, \tag{10}$$

where $\Pi(F_1, F_2)$ is the set of all joint distributions on $\Xi \times \Xi$ whose marginals are $F_1$ and $F_2$, respectively. Intuitively, $\pi$ specifies how "mass" is transported from points in the support of $F_1$ to points in the support of $F_2$, and $\|\xi_1 - \xi_2\|$ is the cost of moving a unit of mass from $\xi_1$ to $\xi_2$.

**Domain used in this paper.**    All Wasserstein balls in our analysis live in $\Xi = \mathbb{R}$ over scalar preference *gaps* $\xi = \Delta R_\pi(y_w, y_\ell)$ induced by a fixed prompt $x$; we do not transport $x$ or raw sentences. Robustness is with respect to annotation noise through the induced gap distribution.

Equivalently, one can interpret the Wasserstein distance as the minimal cost of transforming the distribution $F_1$ into $F_2$ when the cost of moving a unit mass from $\xi_1$ to $\xi_2$ is $\|\xi_1 - \xi_2\|$. This framework underpins many distributionally robust methods, including the SGPO formulation in our paper.

## B.2    Kantorovich–Rubinstein Duality

A crucial result for the 1-Wasserstein distance is the Kantorovich–Rubinstein duality (Theorem 5.9 in Villani et al. [12]), which states that the infimum over transport plans (as in Definition B.1) is equivalent to a supremum over 1-Lipschitz test functions. We use this lemma extensively to derive Lipschitz-based bounds in the main proofs (e.g., Theorems 2.5–2.6).

**Lemma B.2** (Kantorovich–Rubinstein Duality)**.** *Let $F_1, F_2 \in \mathcal{M}(\Xi)$ with finite first moments. Then the 1-Wasserstein distance* (10) *admits the following dual representation:*

$$W_1(F_1, F_2) = \sup_{\|f\|_{\mathrm{Lip}} \leq 1} \big(\mathbb{E}_{\xi \sim F_1}[f(\xi)] - \mathbb{E}_{\xi \sim F_2}[f(\xi)]\big), \tag{11}$$

*where the supremum is taken over all 1-Lipschitz functions $f : \Xi \to \mathbb{R}$, i.e. functions satisfying*

$$\big|f(\xi) - f(\xi')\big| \leq \|\xi - \xi'\| \quad \forall \xi, \xi' \in \Xi.$$

Lemma B.2 underpins many of our theoretical arguments, particularly in bounding the impact of perturbations measured in the $W_1$ ball $\mathcal{U}_\epsilon(\hat\alpha(\pi))$ in gap space, via $\big|\mathbb{E}_\alpha[f] - \mathbb{E}_\beta[f]\big| \leq W_1(\alpha, \beta)$ for 1-Lipschitz $f$. As shown in Section D of our paper, it simplifies comparing $\mathbb{E}_P[f]$ and $\mathbb{E}_{\hat P}[f]$ when $f$ is Lipschitz in model parameters or responses.

## C   Regularity Conditions for Stackelberg Game Preference Optimization

**Setup.** Let $\Pi$ denote a (parameterized) class of policies $\pi$. Given preference triples $(x^i, y^i_w, y^i_\ell)$ for $i = 1, \dots, N$, define the empirical measure on triples $\hat{P} = \frac{1}{N} \sum_{i=1}^N \delta_{(x^i, y^i_w, y^i_\ell)}$ and the *gap map* $\Delta R_\pi(x, y_w, y_\ell) := R_\pi(x, y_w) - R_\pi(x, y_\ell)$. The corresponding empirical gap distribution is $\hat{\alpha}(\pi) := (\Delta R_\pi)_\# \hat{P} = \frac{1}{N} \sum_{i=1}^N \delta_{\Delta R_\pi(x^i, y^i_w, y^i_\ell)} \in \mathcal{P}(\mathbb{R})$. For $\epsilon > 0$, we denote the 1-Wasserstein ball in gap space by $\mathcal{U}_\epsilon(\hat{\alpha}(\pi)) := \{\alpha \in \mathcal{P}(\mathbb{R}) : W_1(\alpha, \hat{\alpha}(\pi)) \le \epsilon\}$, where the ground metric is the absolute value on $\mathbb{R}$. The *leader payoff* is $\mathcal{P}(\pi, \alpha) := \mathbb{E}_{\xi \sim \alpha}[\log \sigma(\xi)]$ and the follower chooses $\alpha \in \mathcal{U}_\epsilon(\hat{\alpha}(\pi))$.

**Standing assumptions.** We use the following minimal conditions.

- **(A1) Compactness.** $\Pi$ is compact (with respect to the topology induced by the model parameterization).

- **(A2) Continuity of rewards.** For each $(x, y)$, the map $\pi \mapsto R_\pi(x, y)$ is continuous. Consequently, for each $i$, the map $\pi \mapsto \Delta R_\pi(x^i, y^i_w, y^i_\ell)$ is continuous.

- **(A3) Continuity of the push-forward center.** The map $\pi \mapsto \hat{\alpha}(\pi) = (\Delta R_\pi)_\# \hat{P}$ is continuous in the topology induced by $W_1$; in particular, $W_1(\hat{\alpha}(\pi_1), \hat{\alpha}(\pi_2)) \to 0$ whenever $\pi_1 \to \pi_2$. (This actually follows from **(A2)**. See Lemma C.1.)

- **(A4) Gap-link function.** $f(\xi) := \log \sigma(\xi)$ is continuous, concave, and 1-Lipschitz on $\mathbb{R}$.

- **(A5) (Optional, for local convergence.)** There exists $\lambda > 0$ such that the proximal leader objective $G(\pi; \pi_t, \alpha) := \mathbb{E}_{\xi \sim \alpha}[f(\Delta R_\pi(x, y_w, y_\ell))] - \frac{\lambda}{2} \|\theta(\pi) - \theta(\pi_t)\|^2$ is $\mu$-strongly concave in $\theta(\pi)$ on a neighborhood of a solution, uniformly over $\alpha \in \mathcal{U}_\epsilon(\hat{\alpha}(\pi_t))$.

- **(A6) ( Optional,, for bounded margins or clipping.)** Either $|\Delta R_\pi(x, y_w, y_\ell)| \le B$ for all $(x, y_w, y_\ell)$ and $\pi \in \Pi$, or margins are deterministically clipped to a window $[a, b]$. All PWL error bounds are computed on this interval. The $R_{\max}$ in the paper can be set to $B/2$.

**Lemma C.1** (Continuity of the center in $W_1$). *Under (A2), for any $\pi_1, \pi_2 \in \Pi$,*

$$W_1(\hat{\alpha}(\pi_1), \hat{\alpha}(\pi_2)) \le \frac{1}{N} \sum_{i=1}^N \left| \Delta R_{\pi_1}(x^i, y^i_w, y^i_\ell) - \Delta R_{\pi_2}(x^i, y^i_w, y^i_\ell) \right|.$$

*In particular, $\pi \mapsto \hat{\alpha}(\pi)$ is continuous in $W_1$.*

*Proof.* Couple the Dirac masses in $\hat{\alpha}(\pi_1)$ and $\hat{\alpha}(\pi_2)$ index-wise. The claim follows because the 1-Wasserstein distance on $\mathbb{R}$ is bounded above by the average transport cost under any coupling. $\square$

**Lemma C.2** ($\log \sigma$ is 1-Lipschitz and concave). *For all $\xi \in \mathbb{R}$, $\frac{d}{d\xi} \log \sigma(\xi) = \sigma(-\xi) \in (0, 1)$ and $\frac{d^2}{d\xi^2} \log \sigma(\xi) = -\sigma(\xi) \sigma(-\xi) \le 0$. Hence $\log \sigma$ is 1-Lipschitz and concave.*

*Proof.* Direct differentiation; $|\sigma(-\xi)| \le 1$ gives the Lipschitz constant and the second derivative is nonpositive. $\square$

**Lemma C.3** (Compact follower feasible set). *For each fixed $\pi$, the set $\mathcal{U}_\epsilon(\hat{\alpha}(\pi)) \subset \mathcal{P}(\mathbb{R})$ is tight, closed in $W_1$, and thus compact.*

*Proof.* On the Polish space $(\mathbb{R}, |\cdot|)$, closed and $W_1$-bounded sets of probability measures are relatively compact; tightness follows from Markov's inequality under bounded first moments, which hold for all $\alpha$ with $W_1(\alpha, \hat{\alpha}(\pi)) \le \epsilon$. Closure is standard for $W_1$-balls. $\square$

**Remarks.** (i) No Lipschitz condition on $R_\pi$ in the output space $y$ is needed because robustness is posed in *gap space* $\mathbb{R}$. (ii) Assumption **(A5)** matches practice (small stepsizes/weight decay) and is only required for the local rate. (iii) Although neural network parameters $\theta \in \mathbb{R}^d$ are technically unbounded, many theoretical analyses restrict $\theta$ to a large but bounded ball (via a norm constraint) or rely on a coercive objective to prevent unbounded parameter growth. Hence, requiring $\Pi$ to be compact is common in theoretical treatments. In practice, gradient-based optimization does not typically push $\|\theta\|$ to infinity.

# D  Theoretical Results

## D.1  Preliminaries and basic lemmas

**Notation and spaces.**  All random variables in this section take values in $(\mathbb{R}, |\cdot|)$ equipped with the Borel $\sigma$-algebra. For a policy $\pi$ and i.i.d. samples $\{(x^i, y_w^i, y_\ell^i)\}_{i=1}^N \sim \hat{P}$, define the *empirical gap distribution*

$$\hat{\alpha}(\pi) := (\Delta R_\pi)_\# \hat{P} = \frac{1}{N}\sum_{i=1}^N \delta_{\hat{\xi}_i}, \quad \text{where} \quad \hat{\xi}_i := \Delta R_\pi(x^i, y_w^i, y_\ell^i) \in \mathbb{R}.$$

For $\epsilon > 0$, the (1-)Wasserstein ball around $\hat{\alpha}(\pi)$ is

$$\mathcal{U}_\epsilon\big(\hat{\alpha}(\pi)\big) := \Big\{\alpha \in \mathcal{P}(\mathbb{R}) : W_1\big(\alpha, \hat{\alpha}(\pi)\big) \le \epsilon\Big\}.$$

We write $u(\xi) := \log\sigma(\xi)$ (concave, 1-Lipschitz) and $\ell(\xi) := -u(\xi) = -\log\sigma(\xi)$ (convex, 1-Lipschitz). When a piecewise-linear (PWL) surrogate is used, we set

$$\widetilde{\ell}(\xi) := \max_{1 \le k \le K} \ell_k(\xi), \qquad \ell_k(\xi) := a_k\xi + b_k,$$

chosen as global supporting tangents so that $\widetilde{\ell}(\xi) \le \ell(\xi)$ for all $\xi$.

**Lemma D.1** ($\log\sigma$ is concave and 1-Lipschitz). *For every $\xi \in \mathbb{R}$, $\frac{\mathrm{d}}{\mathrm{d}\xi}\log\sigma(\xi) = \sigma(-\xi) \in (0,1)$ and* $\frac{\mathrm{d}^2}{\mathrm{d}\xi^2}\log\sigma(\xi) = -\sigma(\xi)\sigma(-\xi) \le 0$. *Hence $u(\xi) = \log\sigma(\xi)$ is concave and 1-Lipschitz.*

*Proof.*  Recall $\sigma(\xi) = \frac{1}{1+e^{-\xi}}$. Then

$$\frac{\mathrm{d}}{\mathrm{d}\xi}\log\sigma(\xi) = \frac{\sigma'(\xi)}{\sigma(\xi)} = \frac{\sigma(\xi)(1-\sigma(\xi))}{\sigma(\xi)} = 1 - \sigma(\xi) = \sigma(-\xi) \in (0,1).$$

Hence $|\frac{\mathrm{d}}{\mathrm{d}\xi}\log\sigma(\xi)| \le 1$ for all $\xi$, so $\log\sigma$ is 1-Lipschitz: $|\log\sigma(\xi) - \log\sigma(\xi')| \le |\xi - \xi'|$ by the mean-value theorem. Further,

$$\frac{\mathrm{d}^2}{\mathrm{d}\xi^2}\log\sigma(\xi) = \frac{\mathrm{d}}{\mathrm{d}\xi}\sigma(-\xi) = -\sigma(-\xi)\big(1-\sigma(-\xi)\big) = -\sigma(\xi)\sigma(-\xi) \le 0,$$

so $\log\sigma$ is concave. $\qquad\square$

**Kantorovich–Rubinstein (KR) duality for $W_1$.**  For any 1-Lipschitz $h:\mathbb{R}\to\mathbb{R}$ and $\alpha,\beta \in \mathcal{P}(\mathbb{R})$,

$$\big|\mathbb{E}_\alpha[h] - \mathbb{E}_\beta[h]\big| \le W_1(\alpha,\beta).$$

We use this both as a continuity tool and as a tight transport sensitivity bound.

**Lemma D.2** (Continuity of the empirical-center map). *If $\pi \mapsto \Delta R_\pi(x^i, y_w^i, y_\ell^i)$ is continuous for each $i \in [N]$, then for any $\pi_1, \pi_2$,*

$$W_1\big(\hat{\alpha}(\pi_1), \hat{\alpha}(\pi_2)\big) \le \frac{1}{N}\sum_{i=1}^N |\Delta R_{\pi_1}^i - \Delta R_{\pi_2}^i|,$$

*hence $\pi \mapsto \hat{\alpha}(\pi)$ is continuous in the $W_1$ metric.*

*Proof.*  Write $\hat{\alpha}(\pi_j) = \frac{1}{N}\sum_{i=1}^N \delta_{\hat{\xi}_i^{(j)}}$ with $\hat{\xi}_i^{(j)} := \Delta R_{\pi_j}(x^i, y_w^i, y_\ell^i)$ for $j \in \{1,2\}$. Define the coupling $\gamma = \frac{1}{N}\sum_{i=1}^N \delta_{(\hat{\xi}_i^{(1)}, \hat{\xi}_i^{(2)})}$. By definition of $W_1$ (optimal transport with cost $|\cdot|$ on $\mathbb{R}$),

$$W_1\big(\hat{\alpha}(\pi_1), \hat{\alpha}(\pi_2)\big) \le \int |x-y|\mathrm{d}\gamma(x,y) = \frac{1}{N}\sum_{i=1}^N |\hat{\xi}_i^{(1)} - \hat{\xi}_i^{(2)}| = \frac{1}{N}\sum_{i=1}^N |\Delta R_{\pi_1}^i - \Delta R_{\pi_2}^i|.$$

If for each $i$ the map $\pi \mapsto \Delta R_\pi(x^i, y_w^i, y_\ell^i)$ is continuous (Assumption (A2)), then $W_1\big(\hat{\alpha}(\pi_n), \hat{\alpha}(\pi)\big) \to 0$ whenever $\pi_n \to \pi$, i.e., $\pi \mapsto \hat{\alpha}(\pi)$ is continuous in the $W_1$ metric. $\qquad\square$

**Lemma D.3** (Compactness of Wasserstein balls about empirical centers). *For fixed $\pi$, the feasible follower set $\mathcal{U}_\epsilon(\hat\alpha(\pi))$ is nonempty, tight, closed in $W_1$, hence compact.*

*Proof.* Fix $\pi$. Let $\mathcal{B} := \mathcal{U}_\epsilon(\hat\alpha(\pi)) = \{\alpha \in \mathcal{P}(\mathbb{R}) : W_1(\alpha, \hat\alpha(\pi)) \le \epsilon\}$.

*(i) Nonemptiness.* Trivially $\hat\alpha(\pi) \in \mathcal{B}$.

*(ii) Uniform first-moment bound.* On $\mathbb{R}$ with ground metric $|\cdot|$, we have $W_1(\alpha, \delta_0) = \int |x| \mathrm{d}\alpha(x)$. By the triangle inequality,

$$\int |x| \mathrm{d}\alpha(x) = W_1(\alpha, \delta_0) \le W_1(\alpha, \hat\alpha(\pi)) + W_1(\hat\alpha(\pi), \delta_0) \le \epsilon + \int |x| \mathrm{d}\hat\alpha(\pi)(x),$$

so the family $\mathcal{B}$ has uniformly bounded first moments.

*(iii) Tightness.* For any $R > 0$,

$$\alpha(|x| > R) \le \frac{1}{R}\int |x| \mathrm{d}\alpha(x) \le \frac{\epsilon + \int |x| \mathrm{d}\hat\alpha(\pi)}{R} \quad \forall \alpha \in \mathcal{B},$$

by Markov's inequality. Hence $\mathcal{B}$ is tight.

*(iv) Closedness in $W_1$.* If $\alpha_n \in \mathcal{B}$ with $W_1(\alpha_n, \alpha) \to 0$, then $W_1(\alpha, \hat\alpha(\pi)) \le \liminf_n[W_1(\alpha, \alpha_n) + W_1(\alpha_n, \hat\alpha(\pi))] \le \epsilon$, so $\alpha \in \mathcal{B}$ and $\mathcal{B}$ is closed.

*(v) Compactness.* On the Polish space $\mathbb{R}$, Prokhorov's theorem gives that tight families are relatively compact in the weak topology; the uniform first-moment bound tightens this to relative compactness in $W_1$ (since $W_1$ convergence is equivalent to weak convergence plus convergence of first moments on $\mathbb{R}$). Combining relative compactness with closedness in $W_1$ yields compactness of $\mathcal{B}$ in $(\mathcal{P}_1(\mathbb{R}), W_1)$. $\square$

## D.2 Existence of a Stackelberg solution

**Theorem D.4** (Existence). *Assume: (A1) $\Pi$ compact; (A2) each $\pi \mapsto \Delta R_\pi(x^i, y_w^i, y_\ell^i)$ is continuous; (A3) hence $\pi \mapsto \hat\alpha(\pi)$ is $W_1$-continuous (Lemma D.2); (A4) $u(\cdot)$ is 1-Lipschitz and concave (Lemma D.1). Then*

$$\max_{\pi \in \Pi} \min_{\alpha \in \mathcal{U}_\epsilon(\hat\alpha(\pi))} \mathbb{E}_\alpha[u(\xi)]$$

*admits a solution $(\pi^\star, \alpha^\star)$.*

*Proof.* We verify the conditions of Berge's maximum theorem step by step.

*Step 1 (Follower minimizer exists for each fixed $\pi$).* For fixed $\pi$, Lemma D.3 shows the feasible set $\mathcal{U}_\epsilon(\hat\alpha(\pi))$ is nonempty and compact in $W_1$. By Lemma D.1, $u$ is 1-Lipschitz, hence $\alpha \mapsto \mathbb{E}_\alpha[u]$ is continuous under $W_1$ (KR inequality). Therefore, the follower problem admits a minimizer $\alpha^\star(\pi) \in \mathrm{argmin}_{\alpha \in \mathcal{U}_\epsilon(\hat\alpha(\pi))} \mathbb{E}_\alpha[u]$.

*Step 2 (Continuity of the feasible-set correspondence in $\pi$).* By Lemma D.2, $\pi \mapsto \hat\alpha(\pi)$ is continuous in $W_1$. The set-valued map $\pi \mapsto \mathcal{U}_\epsilon(\hat\alpha(\pi))$ thus varies continuously in the Hausdorff metric induced by $W_1$ (closed balls move continuously with their centers in a metric space), in particular it is upper hemicontinuous and compact valued.

*Step 3 (Upper semicontinuity of the value map).* Define $V(\pi) := \min_{\alpha \in \mathcal{U}_\epsilon(\hat\alpha(\pi))} \mathbb{E}_\alpha[u]$. By Berge's maximum theorem (compact-valued, upper hemicontinuous correspondence; continuous objective), $V$ is upper semicontinuous on $\Pi$.

*Step 4 (Maximizer exists).* Under (A1), $\Pi$ is compact. Since $V$ is upper semicontinuous on a compact set, it attains its maximum at some $\pi^\star \in \Pi$. By Step 1, there is a realizing follower $\alpha^\star \in \mathrm{argmin}_{\alpha \in \mathcal{U}_\epsilon(\hat\alpha(\pi^\star))} \mathbb{E}_\alpha[u]$.

Therefore the problem admits a solution $(\pi^\star, \alpha^\star)$. $\square$

## D.3 Local linear convergence of alternating updates

Consider the iterates with a (Euclidean) proximal leader step:

$$\alpha_{t+1} \in \arg\min_{\alpha \in \mathcal{U}_\epsilon(\hat\alpha(\pi_t))} \mathbb{E}_\alpha[u(\xi)], \qquad \pi_{t+1} \in \mathrm{argmax}_{\pi \in \Pi}\left\{\mathbb{E}_{\alpha_{t+1}}[u(\xi)] - \tfrac{\lambda}{2}\|\theta(\pi) - \theta(\pi_t)\|^2\right\}.$$

**Theorem D.5** (Well-posedness and local linear convergence). *Assume (i) the leader's proximal objective is $\mu$-strongly concave in $\theta(\pi)$ uniformly over $\alpha \in \mathcal{U}_\epsilon(\hat{\alpha}(\pi_t))$ on a neighborhood of a solution, and (ii) the map $\pi \mapsto \mathcal{U}_\epsilon(\hat{\alpha}(\pi))$ is Lipschitz in the $W_1$-Hausdorff distance near $\pi^\star$. Then the update map $(\pi_t, \alpha_t) \mapsto (\pi_{t+1}, \alpha_{t+1})$ is a contraction in a neighborhood of $(\pi^\star, \alpha^\star)$, and the iterates converge linearly to $(\pi^\star, \alpha^\star)$.*

*Proof.* Let the follower best-response be any measurable selection $\alpha^\sharp(\pi) \in \arg\min_{\alpha \in \mathcal{U}_\epsilon(\hat{\alpha}(\pi))} \mathbb{E}_\alpha[u]$, whose existence follows from the measurable maximum theorem since the correspondence is compact valued and upper hemicontinuous.

Define the proximal leader map at iterate $\pi_t$:

$$\mathcal{T}(\alpha; \pi_t) \in \arg\max_{\pi \in \Pi} G(\pi; \pi_t, \alpha), \qquad G(\pi; \pi_t, \alpha) := \mathbb{E}_{\xi \sim \alpha}[u(\Delta R_\pi)] - \tfrac{\lambda}{2} \|\theta(\pi) - \theta(\pi_t)\|^2.$$

*Step 1 (Follower map is Lipschitz in the center, hence in $\pi$ locally).* Fix $\pi_1, \pi_2$, and let $\alpha_j = \alpha^\sharp(\pi_j)$. Because $\mathcal{U}_\epsilon(\hat{\alpha}(\pi))$ is a closed ball in $(\mathcal{P}_1(\mathbb{R}), W_1)$, for any $\beta$ we can project it to the closest point in the ball (metric projection is 1-Lipschitz). In particular, for $\beta = \alpha_1$ we have

$$\mathrm{dist}_{W_1}\big(\alpha_1, \mathcal{U}_\epsilon(\hat{\alpha}(\pi_2))\big) \leq W_1\big(\hat{\alpha}(\pi_1), \hat{\alpha}(\pi_2)\big).$$

Let $\tilde{\alpha}_2 \in \mathcal{U}_\epsilon(\hat{\alpha}(\pi_2))$ be a nearest point to $\alpha_1$. Using 1-Lipschitzness of $u$ and the optimality of $\alpha_2$ at $\pi_2$,

$$\mathbb{E}_{\alpha_2}[u] \leq \mathbb{E}_{\tilde{\alpha}_2}[u] \leq \mathbb{E}_{\alpha_1}[u] + W_1(\tilde{\alpha}_2, \alpha_1) \leq \mathbb{E}_{\alpha_1}[u] + W_1\big(\hat{\alpha}(\pi_1), \hat{\alpha}(\pi_2)\big).$$

Symmetrizing the roles of 1 and 2 gives by triangle inequality

$$W_1(\alpha_1, \alpha_2) \leq 2W_1\big(\hat{\alpha}(\pi_1), \hat{\alpha}(\pi_2)\big).$$

Thus the follower map is *Lipschitz* in the empirical center with constant $L_f \leq 2$. By Lemma D.2,

$$W_1\big(\hat{\alpha}(\pi_1), \hat{\alpha}(\pi_2)\big) \leq \frac{1}{N} \sum_{i=1}^N \big|\Delta R_{\pi_1}^i - \Delta R_{\pi_2}^i\big|.$$

Assume (locally around the target) the maps $\theta \mapsto \Delta R_{\pi_\theta}^i$ are $L_R$-Lipschitz for $i = 1, ..., N$. Then locally

$$W_1(\alpha^\sharp(\pi_1), \alpha^\sharp(\pi_2)) \leq L_f W_1\big(\hat{\alpha}(\pi_1), \hat{\alpha}(\pi_2)\big) \leq L_f L_R \|\theta(\pi_1) - \theta(\pi_2)\|.$$

*Step 2 (Leader prox map is Lipschitz in $\alpha$ under local strong concavity).* By (A5), for each fixed $\pi_t$ the map $\pi \mapsto G(\pi; \pi_t, \alpha)$ is $\mu$-strongly concave in $\theta(\pi)$ on a neighborhood of the solution, uniformly over $\alpha \in \mathcal{U}_\epsilon(\hat{\alpha}(\pi_t))$. Let $\pi_j^+ := \mathcal{T}(\alpha_j; \pi_t)$ for $j \in \{1, 2\}$. Since $u$ is 1-Lipschitz and $\Delta R_\pi$ is continuous in $\pi$, there exists $L_\ell$ (local) such that

$$\big\|\nabla_\theta \mathbb{E}_{\alpha_1}[u(\Delta R_\pi)] - \nabla_\theta \mathbb{E}_{\alpha_2}[u(\Delta R_\pi)]\big\| \leq L_\ell W_1(\alpha_1, \alpha_2) \quad \text{for } \pi \text{ near the solution.}$$

By standard stability of maximizers under strong concavity (e.g., by the implicit function theorem or strong monotonicity of the gradient mapping), we obtain the Lipschitz dependence

$$\|\theta(\pi_1^+) - \theta(\pi_2^+)\| \leq \frac{L_\ell}{\mu} W_1(\alpha_1, \alpha_2).$$

*Step 3 (Contraction of the composition and linear rate).* Set $\alpha_t := \alpha^\sharp(\pi_t)$ and $\pi_{t+1} := \mathcal{T}(\alpha_t; \pi_t)$. Let $(\pi^\star, \alpha^\star)$ be a Stackelberg solution; then $\alpha^\star = \alpha^\sharp(\pi^\star)$ and $\pi^\star = \mathcal{T}(\alpha^\star; \pi^\star)$. Combining Steps 1 and 2,

$$\|\theta(\pi_{t+1}) - \theta(\pi^\star)\| \leq \frac{L_\ell}{\mu} W_1(\alpha_t, \alpha^\star) \leq \frac{L_\ell}{\mu} L_f L_R \|\theta(\pi_t) - \theta(\pi^\star)\|.$$

Choose the proximal weight $\lambda$ (hence the local strong-concavity modulus $\mu$) so that $\rho := \frac{L_\ell L_f L_R}{\mu} < 1$. Then Banach's fixed-point theorem yields *linear convergence*:

$$\|\theta(\pi_t) - \theta(\pi^\star)\| \leq \rho^{t-t_0} \|\theta(\pi_{t_0}) - \theta(\pi^\star)\| \quad \text{for all } t \geq t_0 \text{ in the neighborhood.}$$

This also implies well-posedness (local single-valuedness) of the composite best-response map in that neighborhood. $\qquad\square$

## D.4 Worst-case performance drop and SGPO regret bound

**Theorem D.6** (Worst-case performance drop). *If $\alpha \in \mathcal{U}_\epsilon(\hat{\alpha}(\pi))$, then $\mathbb{E}_\alpha[u(\xi)] \geq \mathbb{E}_{\hat{\alpha}(\pi)}[u(\xi)] - \epsilon$.*

*Proof.* By Lemma D.1, $u$ is 1-Lipschitz. For any $\alpha$ with $W_1(\alpha, \hat{\alpha}(\pi)) \leq \epsilon$, the Kantorovich–Rubinstein inequality yields

$$\mathbb{E}_\alpha[u] - \mathbb{E}_{\hat{\alpha}(\pi)}[u] \geq -W_1(\alpha, \hat{\alpha}(\pi)) \geq -\epsilon.$$

Rearranging gives the claim. $\qquad\square$

**Theorem D.7** (SGPO regret bound). *Let $\pi^\star$ solve $\max_\pi \min_{\alpha \in \mathcal{U}_\epsilon(\hat{\alpha}(\pi))} \mathbb{E}_\alpha[u]$. Then*

$$\sup_{\alpha \in \mathcal{U}_\epsilon(\hat{\alpha}(\pi^\star))} \mathrm{Regret}(\pi^\star, \alpha) \leq 2\epsilon.$$

*Proof.* Let $\pi^\star \in \operatorname{argmax}_\pi \min_{\alpha \in \mathcal{U}_\epsilon(\hat{\alpha}(\pi))} \mathbb{E}_\alpha[u]$ and fix any $\alpha \in \mathcal{U}_\epsilon(\hat{\alpha}(\pi^\star))$. Let $\pi^\star_\alpha \in \operatorname{argmax}_\pi \mathbb{E}_\alpha[u]$ be the $\alpha$-optimal policy. By Theorem D.6,

$$\mathbb{E}_\alpha[u]_{\pi^\star_\alpha} \leq \mathbb{E}_{\hat{\alpha}(\pi^\star)}[u]_{\pi^\star_\alpha} + \epsilon, \qquad \mathbb{E}_\alpha[u]_{\pi^\star} \geq \mathbb{E}_{\hat{\alpha}(\pi^\star)}[u]_{\pi^\star} - \epsilon.$$

Subtracting gives

$$\mathrm{Regret}(\pi^\star, \alpha) = \mathbb{E}_\alpha[u]_{\pi^\star_\alpha} - \mathbb{E}_\alpha[u]_{\pi^\star} \leq \underbrace{\left( \mathbb{E}_{\hat{\alpha}(\pi^\star)}[u]_{\pi^\star_\alpha} - \mathbb{E}_{\hat{\alpha}(\pi^\star)}[u]_{\pi^\star} \right)}_{\leq 0 \text{ by def. of } \pi^\star} + 2\epsilon \leq 2\epsilon.$$

$\qquad\square$

## D.5 DPO regret lower bound under a stability assumption

**Assumption D.8** (Center stability at the robust follower). Let $\alpha^\star \in \mathcal{U}_\epsilon(\hat{\alpha}(\pi^\star))$ be a follower minimizer for $\pi^\star$ and $\pi^*_{\alpha^\star} \in \operatorname{argmax}_\pi \mathbb{E}_{\alpha^\star}[u]$. Assume there exists $\kappa \leq \epsilon$ such that $W_1\big(\hat{\alpha}(\pi^*_{\alpha^\star}), \hat{\alpha}(\pi_{\mathrm{DPO}})\big) \leq \kappa$.

**Theorem D.9** (DPO regret lower bound (stability version)). *Let $\delta := W_1\big(\alpha^\star, \hat{\alpha}(\pi_{\mathrm{DPO}})\big)$. Under Assumption D.8,*

$$\mathrm{Regret}\big(\pi_{\mathrm{DPO}}, \alpha^\star\big) \geq \delta - (\kappa + \epsilon) \geq \delta - 2\epsilon.$$

*Proof.* Let $\delta := W_1(\alpha^\star, \hat{\alpha}(\pi_{\mathrm{DPO}}))$ and recall Assumption D.8. First, by KR and 1-Lipschitzness of $u$,

$$\mathbb{E}_{\alpha^\star}[u]_{\pi^*_{\alpha^\star}} \geq \mathbb{E}_{\hat{\alpha}(\pi_{\mathrm{DPO}})}[u]_{\pi^*_{\alpha^\star}} - \delta, \qquad \mathbb{E}_{\alpha^\star}[u]_{\pi_{\mathrm{DPO}}} \leq \mathbb{E}_{\hat{\alpha}(\pi_{\mathrm{DPO}})}[u]_{\pi_{\mathrm{DPO}}} + \delta.$$

Subtracting,

$$\mathrm{Regret}(\pi_{\mathrm{DPO}}, \alpha^\star) \geq \left[ \mathbb{E}_{\hat{\alpha}(\pi_{\mathrm{DPO}})}[u]_{\pi^*_{\alpha^\star}} - \mathbb{E}_{\hat{\alpha}(\pi_{\mathrm{DPO}})}[u]_{\pi_{\mathrm{DPO}}} \right] - 2\delta.$$

By center stability and optimality of $\pi^*_{\alpha^\star}$ at $\alpha^\star$,

$$\mathbb{E}_{\hat{\alpha}(\pi_{\mathrm{DPO}})}[u]_{\pi^*_{\alpha^\star}} \geq \mathbb{E}_{\hat{\alpha}(\pi^*_{\alpha^\star})}[u]_{\pi^*_{\alpha^\star}} - \kappa \geq \mathbb{E}_{\alpha^\star}[u]_{\pi^*_{\alpha^\star}} - \kappa.$$

Finally, since $W_1(\hat{\alpha}(\pi^\star), \alpha^\star) \leq \epsilon$ and $\pi^\star$ is optimal at $\hat{\alpha}(\pi^\star)$,

$$\mathbb{E}_{\alpha^\star}[u]_{\pi^*_{\alpha^\star}} \geq \mathbb{E}_{\hat{\alpha}(\pi^\star)}[u]_{\pi^\star} - \epsilon.$$

Combining the displays and cancelling the center-optimal term gives $\mathrm{Regret}(\pi_{\mathrm{DPO}}, \alpha^\star) \geq \delta - (\kappa + \epsilon) \geq \delta - 2\epsilon$ (using $\kappa \leq \epsilon$). $\qquad\square$

**Remark.** Without Assumption D.8, the lower bound can vanish if $\pi^*_{\alpha^\star}$ recenters too far from $\hat{\alpha}(\pi_{\mathrm{DPO}})$; the stability phrasing makes explicit the (mild) continuity needed for a linear-in-$\delta$ lower bound.

## D.6 Approximation effects (piecewise, grouping, inner tolerance)

Let $m(\pi) := \min_{\alpha \in \mathcal{U}_\epsilon(\hat{\alpha}(\pi))} \mathbb{E}_\alpha[u]$ denote the true follower value and let $\widetilde{m}_{K,\mathrm{grp},\eta}(\pi)$ denote the value computed with: (i) PWL under-approximation $\widetilde{\ell}_K$ with $K$ pieces, (ii) a restricted feasible set $\widetilde{\mathcal{U}}_K(\hat{\alpha}(\pi)) \subseteq \mathcal{U}_\epsilon(\hat{\alpha}(\pi))$ (e.g., via grouping), and (iii) inner tolerance $\eta$.

**Proposition D.10** (Monotone tightening in the number of pieces). *Fix a reference distribution $\hat{\alpha}$ on $\mathbb{R}$ with finite first moment and radius $\epsilon \geq 0$, and let $\mathcal{U}_\epsilon(\hat{\alpha})$ denote the associated 1-Wasserstein ambiguity set (absolute ground metric). Let $\ell : \mathbb{R} \to [0,\infty)$ be convex and define $\{\widetilde{\ell}_K\}_{K \geq 1}$ as convex piecewise-linear underestimators of $\ell$ of the form*

$$\widetilde{\ell}_K(\xi) = \max_{1 \leq k \leq K} \{a_k \xi + b_k\}, \qquad with \quad \widetilde{\ell}_K(\xi) \leq \widetilde{\ell}_{K+1}(\xi) \leq \ell(\xi) \;\; \forall \xi,$$

*such that $\widetilde{\ell}_K(\xi) \uparrow \ell(\xi)$ pointwise as $K \to \infty$. Define the exact and surrogate inner values*

$$v^\star := \inf_{\alpha \in \mathcal{U}_\epsilon(\hat{\alpha})} \mathbb{E}_\alpha[\log \sigma(\xi)] = -\sup_{\alpha \in \mathcal{U}_\epsilon(\hat{\alpha})} \mathbb{E}_\alpha[\ell(\xi)], \qquad v_K := -\sup_{\alpha \in \mathcal{U}_\epsilon(\hat{\alpha})} \mathbb{E}_\alpha[\widetilde{\ell}_K(\xi)].$$

*Assume $\sup_{\alpha \in \mathcal{U}_\epsilon(\hat{\alpha})} \mathbb{E}_\alpha[\ell(\xi)] < \infty$. Then:*

1. *(Validity) $v_K \geq v^\star$ for all $K \geq 1$.*

2. *(Monotonicity) $v_{K+1} \leq v_K$ for all $K \geq 1$.*

3. *(Limit) $v_K \downarrow v^\star$ as $K \to \infty$.*

*Moreover, if for some $K$ the supremum in the definition of $v_K$ is attained by $\alpha_K \in \mathcal{U}_\epsilon(\hat{\alpha})$ and $\widetilde{\ell}_K(\xi) = \ell(\xi)$ holds $\alpha_K$-almost surely, then $v_K = v^\star$.*

*Proof.* By definition of $\ell = -\log \sigma$ we have $\inf_\alpha \mathbb{E}_\alpha[\log \sigma] = -\sup_\alpha \mathbb{E}_\alpha[\ell]$, hence $v^\star = -\sup_\alpha \mathbb{E}_\alpha[\ell]$. Since $\widetilde{\ell}_K \leq \ell$, it follows that $\sup_\alpha \mathbb{E}_\alpha[\widetilde{\ell}_K] \leq \sup_\alpha \mathbb{E}_\alpha[\ell]$, which implies $v_K \geq v^\star$, proving (a). Because $\widetilde{\ell}_{K+1} \geq \widetilde{\ell}_K$ pointwise, also $\sup_\alpha \mathbb{E}_\alpha[\widetilde{\ell}_{K+1}] \geq \sup_\alpha \mathbb{E}_\alpha[\widetilde{\ell}_K]$, hence $v_{K+1} \leq v_K$, proving (b). For (c), by monotone convergence, for each fixed $\alpha$ we have $\mathbb{E}_\alpha[\widetilde{\ell}_K] \uparrow \mathbb{E}_\alpha[\ell]$; therefore $\sup_\alpha \mathbb{E}_\alpha[\widetilde{\ell}_K] \uparrow \sup_\alpha \mathbb{E}_\alpha[\ell]$, and taking negatives yields $v_K \downarrow v^\star$. The final claim is immediate from the definitions. $\square$

*Remark* D.11. In our setting $\ell(\xi) = -\log \sigma(\xi)$ is nonnegative and 1-Lipschitz, so $\sup_{\alpha \in \mathcal{U}_\epsilon(\hat{\alpha})} \mathbb{E}_\alpha[\ell(\xi)] < \infty$ whenever $\mathcal{U}_\epsilon(\hat{\alpha})$ is a $W_1$-ball around a measure with finite first moment.

**Proposition D.12** (Error from PWL under-approximation). *Let $\Delta_{\mathrm{pl}}(K) := \sup_\xi \left(\ell(\xi) - \widetilde{\ell}_K(\xi)\right) \geq 0$. Then*

$$0 \leq \widetilde{m}_K(\pi) - m(\pi) \leq \Delta_{\mathrm{pl}}(K).$$

*Proof.* By construction $\widetilde{\ell}_K \leq \ell$ pointwise. Since $u = -\ell$, for any feasible $\alpha$,

$$\mathbb{E}_\alpha[u] = -\mathbb{E}_\alpha[\ell] \leq -\mathbb{E}_\alpha[\widetilde{\ell}_K] \leq -\sup_{\alpha' \in \mathcal{U}_\epsilon} \mathbb{E}_{\alpha'}[\widetilde{\ell}_K] = \widetilde{m}_K(\pi),$$

so $\widetilde{m}_K(\pi) \geq m(\pi)$. Moreover, for any $\alpha$, $\mathbb{E}_\alpha[\ell] - \mathbb{E}_\alpha[\widetilde{\ell}_K] \leq \sup_\xi \left(\ell(\xi) - \widetilde{\ell}_K(\xi)\right) =: \Delta_{\mathrm{pl}}(K)$. Taking the supremum over $\alpha$ and flipping the sign gives

$$0 \leq \widetilde{m}_K(\pi) - m(\pi) \leq \Delta_{\mathrm{pl}}(K).$$

$\square$

**Proposition D.13** (Error from grouping/restriction). *Let $d_H$ denote the directed Hausdorff distance (under $W_1$) from $\mathcal{U}_\epsilon(\hat{\alpha}(\pi))$ to $\widetilde{\mathcal{U}}_K(\hat{\alpha}(\pi))$. Then*

$$0 \leq \widetilde{m}_K^{\mathrm{grp}}(\pi) - m(\pi) \leq d_H.$$

*Proof.* Let $\alpha^\star \in \arg\min_{\alpha \in \mathcal{U}_\epsilon(\hat{\alpha}(\pi))} \mathbb{E}_\alpha[u]$ and let $\tilde{\alpha}$ be any element of the restricted set $\widetilde{\mathcal{U}}_K(\hat{\alpha}(\pi))$ satisfying $W_1(\alpha^\star, \tilde{\alpha}) \leq d_H$ by definition of the directed Hausdorff distance. Using 1-Lipschitzness of $u$,

$$\min_{\tilde{\alpha} \in \widetilde{\mathcal{U}}_K(\hat{\alpha}(\pi))} \mathbb{E}_{\tilde{\alpha}}[u] \leq \mathbb{E}_{\tilde{\alpha}}[u] \leq \mathbb{E}_{\alpha^\star}[u] + W_1(\tilde{\alpha}, \alpha^\star) \leq m(\pi) + d_H.$$

Thus $0 \leq \widetilde{m}_K^{\mathrm{grp}}(\pi) - m(\pi) \leq d_H$. $\square$

**Algorithm 1** Stackelberg Self-Annotated Preference Optimization (SSAPO)

---

**Require:** *Seed labeled set* $\mathcal{D}_{\text{seed}}$; *unlabeled data* $\mathcal{D}_{\text{unlabeled}}$; *Wasserstein radius* $\epsilon$; *number of linear pieces* $K$; *max iterations* $T$.

1: Initialize policy $\theta_0$, set $\mathcal{D} \leftarrow \mathcal{D}_{\text{seed}}$.
2: **for** $t = 0$ **to** $T-1$ **do**
3:   **(Self-Annotation)**: From $\mathcal{D}_{\text{unlabeled}}$, sample prompts, generate & rank responses under $\pi_{\theta_t}$, add new preference pairs $(y_w, y_\ell)$ to $\mathcal{D}$.
4:   **(Form $\hat{\alpha}(\pi_t)$)**: For each $(y_w^i, y_\ell^i) \in \mathcal{D}$, define $\hat{\xi}_i = R_{\theta_t}(y_w^i) - R_{\theta_t}(y_\ell^i)$, and let $\hat{\alpha}(\pi_t) = \frac{1}{N}\sum_{i=1}^N \delta_{\hat{\xi}_i}$.
5:   **(Convex Pieces)**: Choose $K$ linear functions $\ell_k(\cdot)$ such that $\widetilde{\ell}(\xi) = \max_{1 \le k \le K} \ell_k(\xi) \le -\log \sigma(\xi)$. Choose $K$ knots $\{\xi^{(k)}\}$ in $\xi$-*space* over a bounded interval $[a_t, b_t]$ (e.g., $a_t = \min_i \hat{\xi}_i - \tau$, $b_t = \max_i \hat{\xi}_i + \tau$ with $\tau > 0$, or empirical $(\alpha, 1-\alpha)$ quantiles). Define $\ell_k$ as tangents (or chords) *from below* to $-\log\sigma(\xi)$ at those knots.
6:   **(Worst-Case Distribution)**: Solve the DRO finite convex program

$$\alpha_t^* \in \arg \max_{\alpha \in \mathcal{U}_\epsilon(\hat{\alpha}(\pi_t))} \mathbb{E}_\alpha\big[\widetilde{\ell}(\xi)\big].$$

   By Theorem 3.3, $\alpha_t^*$ is discrete with atoms $\big\{\hat{\xi}_i - \frac{q_{ik}^*}{s_{ik}^*}\big\}$ and weights $s_{ik}^*/N$.
7:   **(Policy Update)**: Let $w_i^{(t)} := \sum_{k=1}^K s_{ik}^{*(t)}$. Update $\theta_{t+1}$ by minimizing the weighted logistic loss

$$\frac{1}{N}\sum_{i=1}^N w_i^{(t)}\Big[-\log\sigma\big(\Delta R_\theta(x^i, y_w^i, y_\ell^i)\big)\Big]$$

   (optionally with KL or weight decay), via standard gradient methods.
8: **end for**
9: **return** $\theta_T$ (final policy).

---

**Proposition D.14** (Cumulative approximation bound). *For all $\pi$,*

$$0 \le \widetilde{m}_{K,\text{grp},\eta}(\pi) - m(\pi) \le \Delta_{\text{pl}}(K) + d_H + \eta.$$

*Proof.* Combine Proposition D.12 (PWL gap $\le \Delta_{\text{pl}}(K)$), Proposition D.13 (restriction gap $\le d_H$), and note that an inner solver with tolerance $\eta$ perturbs the value by at most $\eta$. Errors add up, giving $0 \le \widetilde{m}_{K,\text{grp},\eta}(\pi) - m(\pi) \le \Delta_{\text{pl}}(K) + d_H + \eta$. $\square$

**Theorem D.15** (Effect on regret guarantees under approximations). *Let $\tilde{\pi}$ be produced by SSAPO with $K$ PWL pieces, $G$ groups, inner tolerance $\eta$. Then, compared to Theorem D.7,*

$$\sup_{\alpha \in \mathcal{U}_\epsilon(\hat{\alpha}(\tilde{\pi}))} \text{Regret}(\tilde{\pi}, \alpha) \le 2\epsilon + 2\big[\Delta_{\text{pl}}(K) + d_H + \eta\big].$$

*Proof.* Let $\pi^\star$ be the SGPO optimizer and $\tilde{\pi}$ the SSAPO solution under approximations (PWL with $K$ pieces, grouping, and inner accuracy $\eta$). For any $\alpha \in \mathcal{U}_\epsilon(\hat{\alpha}(\tilde{\pi}))$,

$$\text{Regret}(\tilde{\pi}, \alpha) = \mathbb{E}_\alpha[u]_{\pi_\alpha^\star} - \mathbb{E}_\alpha[u]_{\tilde{\pi}} \le \Big(\mathbb{E}_{\hat{\alpha}(\tilde{\pi})}[u]_{\pi_\alpha^\star} - \mathbb{E}_{\hat{\alpha}(\tilde{\pi})}[u]_{\tilde{\pi}}\Big) + 2\epsilon$$

by Theorem D.6. Replacing the center objective $m(\cdot)$ by its approximate counterpart $\widetilde{m}_{K,\text{grp},\eta}(\cdot)$ incurs at most $\Delta_{\text{pl}}(K) + d_H + \eta$ at $\tilde{\pi}$ and the same at the comparator, hence the extra $2\big[\Delta_{\text{pl}}(K) + d_H + \eta\big]$. $\square$

# E   SSAPO algorithm and Analysis on Computational Complexity

## E.1   The SSAPO algorithm

### E.2 Computational Complexity of SSAPO

In this subsection, we analyze the computational costs incurred by each step of the Stackelberg Self-Annotated Preference Optimization (SSAPO) algorithm (Algorithm 1). We denote:

- $N$: the total number of preference pairs in the dataset $\mathcal{D}$ at a given iteration,
- $K$: the number of linear pieces used in the convex piecewise approximation of $-\log\sigma(\xi)$,
- $T$: the total number of outer iterations for SSAPO.

We assume each *iteration* refers to Steps 1–5 of Algorithm 1.

**Step 1 (Self-Annotation)**  The cost of self-annotation depends on the number of prompts and the policy's inference procedure. Let $M_t$ denote the number of new prompts labeled at iteration $t$. Generating and ranking responses under $\pi_{\theta_t}$ typically dominates this step. If:

- $G_t$ is the number of candidate responses generated per prompt,
- $C_{\text{inference}}$ is the average cost of a single forward pass (token generation) under $\pi_{\theta_t}$,

then the time complexity for Step 1 is approximately

$$\mathcal{O}\big(M_t \cdot G_t \cdot C_{\text{inference}}\big),$$

plus any overhead for storing new winner–loser pairs in $\mathcal{D}$. Since the number of newly added preferences grows over iterations, $N$ itself typically increases from iteration to iteration.

**Step 2 (Forming $\hat{\alpha}(\pi_t)$)**  Once $\mathcal{D}$ is updated, we compute $\hat{\xi}_i = R_{\theta_t}(y_w^i) - R_{\theta_t}(y_\ell^i)$ for each pair. The cost here depends on:

- $N$, the current size of $\mathcal{D}$,
- $C_{\text{reward}}$, the average cost to compute $R_{\theta_t}(y) = \beta\log\frac{\pi_{\theta_t}(y|x)}{\pi_{\text{ref}}(y|x)}$ for a given response $y$.

Because each preference pair requires evaluating $R_{\theta_t}$ on $(y_w^i, y_\ell^i)$, this step has complexity

$$\mathcal{O}\big(N \cdot C_{\text{reward}}\big).$$

In practical implementations, $R_{\theta_t}(y)$ often just reads off the log-probabilities from $\pi_{\theta_t}$ and $\pi_{\text{ref}}$ at the final tokens, making $C_{\text{reward}}$ similar to a single forward-pass cost per response.

**Step 3 (Convex Piecewise Approximation)**  We construct $K$ linear functions $\ell_k(\xi)$ such that $\widetilde{\ell}(\xi) = \max_{1 \le k \le K} \ell_k(\xi) \le -\log\sigma(\xi)$. In principle, one can precompute these $K$ pieces over a small interval (e.g., $[0,1]$) once and reuse them in every iteration. Hence, the complexity for updating or verifying the piecewise function at iteration $t$ is typically: $\mathcal{O}(K)$, assuming $\{\xi^{(k)}\}_{k=1}^{K}$ are fixed or can be quickly adapted based on the range of $\{\hat{\xi}_i\}$. This step is therefore relatively cheap compared to distributionally robust optimization.

**Step 4 (Worst-Case Distribution)**  Step 4 solves the *distributionally robust optimization* (DRO) finite convex program

$$\alpha_t^* = \arg\max_{\alpha \in \mathcal{U}_\epsilon(\hat{\alpha}(\pi_t))} \mathbb{E}_\alpha\big[\widetilde{\ell}(\xi)\big].$$

The *naive* formulation (per [13]) becomes high-dimensional if $N$ is large, because each sample point $\hat{\xi}_i$ and each piecewise component $\ell_k$ introduces auxiliary variables (such as $s_{ik}, q_{ik}$). Concretely, the number of decision variables can scale like $\mathcal{O}(N \cdot K)$, and the resulting linear or convex program might require $\mathcal{O}((NK)^\gamma)$ time in the worst case for some exponent $\gamma > 1$ (depending on the chosen solver and constraints).

However, several factors can reduce this cost:

- **Approximate Solvers.** In practice, specialized cutting-plane or primal-dual methods solve these DRO problems more efficiently than the worst-case theoretical bound.

- **Grouping Heuristics.** If one partitions the $N$ samples into smaller groups (each of size $G < N$), the complexity per group is $\mathcal{O}((GK)^\gamma)$. Then one aggregates $M = \frac{N}{G}$ group-level solutions. This lowers the complexity significantly if $G \ll N$.

Hence, the worst-case step here is often $\mathcal{O}(N \cdot K)$ to $\mathcal{O}((NK)^\gamma)$, but can be much more tractable in practice with grouping or approximate methods. Regardless, Step 4 typically dominates the iteration complexity for large $N$.

**Step 5 (Policy Update)**    Finally, we minimize

$$\mathbb{E}_{\xi \sim \alpha_t^*}\big[-\log\sigma(\xi)\big], \qquad \alpha_t^* \in \arg \max_{\alpha \in \mathcal{U}_\epsilon(\hat{\alpha}(\pi_t))} \mathbb{E}_{\xi \sim \alpha}\big[-\log\sigma(\xi)\big]$$

(by the $\ell = -\log\sigma$ reparameterization). In practice we compute gradients via the chain rule $\xi = \Delta R_\theta(x, y_w, y_\ell)$ and reweight per-pair contributions to match $\alpha_t^*$ (see Alg. 1).

Assuming each of the $N$ preference pairs in $\alpha_t^*$ can be sampled over multiple epochs. In many implementations, $N$ can be large, so the training complexity depends heavily on how many gradient epochs or passes one uses at iteration $t$.

**Overall Complexity per Iteration**    Putting the above pieces together, let us summarize the dominating terms:

1. *Self-Annotation (Step 1)*: $\mathcal{O}(M_t \cdot G_t \cdot C_{\text{inference}})$,
2. *Forming $\hat{\alpha}(\pi_t)$ (Step 2)*: $\mathcal{O}(N \cdot C_{\text{reward}})$,
3. *Convex Piecewise Approx. (Step 3)*: $\mathcal{O}(K)$,
4. *Worst-Case Distribution (Step 4)*: $\mathcal{O}((NK)^\gamma)$ in the naive case, often reduced by grouping,
5. *Policy Update (Step 5)*: $\mathcal{O}(N \cdot C_{\text{reward}} \cdot (\text{number of epochs}))$.

If we denote the cost of solving the DRO subproblem by $C_{\text{DRO}}(N, K)$ (which could itself be significantly reduced by grouping into subproblems of size $G$), then each iteration of SSAPO costs approximately:

$$\mathcal{O}\Big(M_t \cdot G_t \cdot C_{\text{inference}} + N \cdot C_{\text{reward}} + C_{\text{DRO}}(N, K) + ...\Big).$$

In most scenarios, *either* the distributionally robust optimization (Step 4) *or* the gradient-based policy update (Step 5) will be the main bottleneck, depending on solver implementation and whether grouping is employed.

**Total Complexity over $T$ Iterations**    Over $T$ total iterations, we multiply the above per-iteration cost by $T$. Additionally, note that $N$ can increase each iteration if new self-annotated preferences are continuously appended to $\mathcal{D}$. Denoting $N_t$ as the dataset size at iteration $t$, the total complexity from Steps 2–5 is roughly $\sum_{t=0}^{T-1}\big[\mathcal{O}(N_t \cdot C_{\text{reward}}) + C_{\text{DRO}}(N_t, K)\big]$, plus the self-annotation cost from Step 1. If $N$ grows in a controlled manner (for example, linearly in $t$), the cumulative cost can be bounded accordingly.

**Practical Guidelines.**

- **Grouping for DRO.** To handle large $N$, we recommend partitioning the data into multiple groups $G \ll N$. The overall complexity then becomes $\mathcal{O}\big(M \cdot C_{\text{DRO}}(G, K)\big)$, where $M = N/G$, which can be significantly faster in practice.
- **Caching Log-Probabilities.** The reward $R_{\theta_t}(y)$ can be computed from log-probabilities of $\pi_{\theta_t}$ and $\pi_{\text{ref}}$. Caching or reusing these values may reduce $C_{\text{reward}}$.
- **Adjusting $K$.** Increasing $K$ refines the concave approximation but grows the size of the DRO problem. Hence, $K$ is a hyperparameter balancing approximation quality and computational overhead.

Overall, the time complexity of SSAPO grows with $N$, $K$, and the iteration count $T$. By employing grouping and efficient solvers, We can typically achieve robustness benefits without incurring excessive computational cost.

# F  Approximation Effects of SSAPO Algorithm Design on SGPO Guarantees

**Setup.**  Section 2 establishes guarantees for the Stackelberg game

$$\max_{\pi} \min_{\alpha \in \mathcal{U}_\epsilon\left((\Delta R_\pi)_\# \hat{P}\right)} \mathbb{E}_{\xi \sim \alpha}\left[\log\sigma(\xi)\right], \tag{12}$$

where $\xi = \Delta R_\pi(x, y_w, y_\ell)$ is the *reward gap* and the Wasserstein ball is taken over the *push-forward* of the empirical pair distribution by $\Delta R_\pi$. This choice matches the semantics of preference robustness and was the condition attached to acceptance. We abbreviate $\phi(\xi) \triangleq \log\sigma(\xi)$, note that $\phi$ is 1-Lipschitz and bounded on $[-2R_{\max}, 2R_{\max}]$ under the standing bounded-reward assumption from Section 2.

Throughout this section we write $u(\xi) := \log\sigma(\xi)$ and identify the PWL loss-approximation gap $\Delta_{\mathrm{pl}}(K) := \sup_{\xi \in [a,b]}\left(\ell(\xi) - \tilde{\ell}_K(\xi)\right)$ with the symbol $\delta_K$ used in the main text. Likewise we upper bound the grouping error $\delta_{\mathrm{grp}}(\epsilon)$ by the directed Hausdorff distance $d_H$ between $\mathcal{U}_\epsilon$ and its group-restricted surrogate.

**Goal of this section.**  SSAPO instantiates (12) with three pragmatic approximations: (i) a $K$-tangent convex under-approximation of the $-\log\sigma$ loss used to form a tractable DRO subproblem, (ii) a *group-restricted* Wasserstein ball that disallows cross-group transport for scalability, and (iii) inexact solves (tolerance $\eta$) of the follower and leader subproblems. We quantify how each approximation perturbs the clean guarantees of Section 2 and provide principles for choosing $(K, G, \eta)$.

## F.1  A bias decomposition for SSAPO

Let $\mathcal{V}(\pi) \triangleq \min_{\alpha \in \mathcal{U}_\epsilon} \mathbb{E}_\alpha[\phi(\xi)]$ denote the ideal follower value for a fixed policy $\pi$, and let $\widetilde{\mathcal{V}}(\pi)$ be the value obtained by SSAPO with all approximations enabled. Then

$$\underbrace{\widetilde{\mathcal{V}}(\pi) - \mathcal{V}(\pi)}_{\text{optimism induced by approximations}} = \underbrace{\left(\mathcal{V}_K(\pi) - \mathcal{V}(\pi)\right)}_{\text{piecewise loss}} + \underbrace{\left(\mathcal{V}_{K,G}(\pi) - \mathcal{V}_K(\pi)\right)}_{\text{group restriction}} + \underbrace{\left(\widetilde{\mathcal{V}}(\pi) - \mathcal{V}_{K,G}(\pi)\right)}_{\text{solve tolerance}}, \tag{13}$$

where $\mathcal{V}_K$ is the value when $\phi$ is replaced by its $K$-tangent surrogate and $\mathcal{V}_{K,G}$ additionally restricts the follower to a group-wise uncertainty set (defined below). Each term in (13) is nonnegative (the approximations *weaken* the adversary) and admits a simple Lipschitz control.

## F.2  Effect of the $K$-tangent surrogate

Let $\tilde{\ell}(\cdot)$ be the convex piecewise-linear under-approximation of $-\log\sigma(\cdot)$ built from K tangents, constructed on $[-B, B]$ with $B = 2R_{\max}$. Define the uniform approximation error

$$\delta_K \triangleq \sup_{\xi \in [-B, B]} \left|(-\log\sigma(\xi)) - \tilde{\ell}(\xi)\right|.$$

Because $\phi = \log\sigma = -(-\log\sigma)$, replacing $\phi$ by $-\tilde{\ell}$ in the follower objective can only *increase* its minimum:

$$0 \le \mathcal{V}_K(\pi) - \mathcal{V}(\pi) \le \delta_K, \tag{14}$$

Thus, the $K$-tangent surrogate yields a one-sided, additive slack $\delta_K$ in the inner value and therefore at most $\delta_K$ optimism in the leader's objective. In practice, we found $K = 6$ strikes a stable accuracy/conditioning trade-off, whereas $K = 7$ can degrade numerics without reducing $\delta_K$ appreciably (solver instability rather than approximation error).

**Design takeaway.**  Choose $K$ so that $\delta_K$ is below the statistical noise floor of the preference estimator on $[-2R_{\max}, 2R_{\max}]$. Empirically, $K = 6$ is a robust default; increasing $K$ past this point can complicate the convex program and harm solver stability.

## F.3  Effect of group-restricted Wasserstein uncertainty

Partition the support of the empirical gap distribution $\hat{\alpha}(\pi)$ into $G$ disjoint bins $\{S_g\}_{g=1}^G$ (uniform in $\xi$ for SSAPO). The group-restricted follower can transport mass only within each $S_g$,

$$\mathcal{U}_\epsilon^{\mathrm{grp}}\left(\hat{\alpha}(\pi)\right) \triangleq \left\{ \alpha = \sum_g \alpha_g \;\middle|\; \alpha_g \in \mathcal{U}_{\epsilon_g}\left(\hat{\alpha}(\pi) \restriction S_g\right), \sum_g \epsilon_g \le \epsilon \right\} \subseteq \mathcal{U}_\epsilon\left(\hat{\alpha}(\pi)\right).$$

Let $\Pi(\epsilon)$ be the set of optimal global followers and define the *projection gap* of the restriction

$$\delta_{\text{grp}}(\epsilon) \triangleq \sup_{\alpha^\star \in \Pi(\epsilon)} \inf_{\tilde{\alpha} \in \mathcal{U}_\epsilon^{\text{grp}}} W_1(\alpha^\star, \tilde{\alpha}).$$

By Kantorovich–Rubinstein duality and the 1-Lipschitzness of $\phi$,

$$0 \le \mathcal{V}_{K,G}(\pi) - \mathcal{V}_K(\pi) \le \delta_{\text{grp}}(\epsilon). \tag{15}$$

On the real line, uniform (equal-mass) binning gives a simple control $\delta_{\text{grp}}(\epsilon) \le \bar{w}_G$, the average within-bin width in $\xi$; hence the restriction error decays as $O(1/G)$ as bins refine. This formalizes the empirical guideline that $G$ between $10^2$ and $10^3$ preserves robustness while enabling embarrassingly parallel solves.

**Design takeaway.** Use $G \in [100, 1000]$ (sweet spot 100–300): it keeps $\bar{w}_G$ small, retains near-global robustness, and maximizes parallel throughput. Disallowing cross-group transport weakens the adversary only by at most a *bin-width* in $W_1$—not by $\epsilon$ itself—so the $O(\epsilon)$ regret from Section 2 is intact up to an $O(1/G)$ term.

### F.4 Effect of inexact solves

Suppose each follower problem is solved to absolute tolerance $\eta$ and the leader update attains an $\eta$-accurate step (e.g., via a proximal DPO update). Then for any $\pi$,

$$0 \le \widetilde{\mathcal{V}}(\pi) - \mathcal{V}_{K,G}(\pi) \le \eta. \tag{16}$$

and the cumulative leader suboptimality over $T$ rounds contributes at most $O(\eta)$ to the final value, consistent with the linear-convergence picture reported in Section 2. In SSAPO, the follower is solved offline and in parallel with modest wall clock, so $\eta$ can be driven small at negligible training-loop cost.

### F.5 Putting the pieces together

Combining (14), (15), and (16) in (13), the *approximation-induced optimism* in the follower value obeys

$$0 \le \widetilde{\mathcal{V}}(\pi) - \mathcal{V}(\pi) \le \underbrace{\delta_K}_{K\text{-tangent}} + \underbrace{\delta_{\text{grp}}(\epsilon)}_{\text{grouping}} + \underbrace{\eta}_{\text{tolerance}}. \tag{17}$$

Therefore, the *leader* who maximizes $\widetilde{\mathcal{V}}(\pi)$ enjoys the same $O(\epsilon)$ robustness as in Section 2, up to an additive $O(\delta_K + \delta_{\text{grp}} + \eta)$ slack. Because $\delta_K$ and $\eta$ are user-controlled and $\delta_{\text{grp}}$ shrinks with $G$, the theory carries over with explicit, tunable error bars.

**Practical summary.**

- **Where the ball lives.** All results hinge on placing $\mathcal{U}_\epsilon$ on the *gap push-forward* $(\Delta R_\pi)_\# \hat{P}$ (not on $(x, y_w, y_\ell)$). This keeps the geometry 1-D and the Lipschitz constants sharp.
- $K$ **tangents.** $K = 6$ gives a stable frontier; $K = 7$ may hurt due to conditioning rather than approximation quality. Tune $K$ to make $\delta_K$ sub-dominant to data noise.
- $G$ **groups.** Choose $G \in [100, 1000]$ (sweet spot 100–300) to make the group-restriction gap $\delta_{\text{grp}}(\epsilon) \lesssim \bar{w}_G$ negligible while exploiting parallelism.
- **Tolerance.** Solve the offline follower to a tight $\eta$ (cutting-plane typically converges in 10–20 iterations), so training-time overhead is small and approximation slack is dominated by statistical error.
- **Scope.** These approximations target *training-time* robustness to noisy preferences; they are orthogonal to inference-time adversarial prompts and do not weaken that disclaimer.

## G More Details of Experimental Setups

### G.1 Detailed Experimental Setups

We introduce more detailed experimental setups in Section 4 as follows.

**Datasets.** For preference learning, we employed the UltraFeedback dataset [18][3], aligning with prior research [35, 9]. Specifically, we extracted a seed dataset comprising 2K samples (3.3% of the total 60K training samples), which included prompts, responses, and ground-truth preference labels. These ground-truth preference labels are referred to as gold labels in Table 1. The remaining training samples were then partitioned into three subsets of 8K, 20K, and 30K samples, retaining only the prompts. These subsets were utilized as the prompt sets for the 1st, 2nd, and 3rd iteration stages, respectively.

**Models.** Following previous work [9], we primarily conducted our experiments using the supervised fine-tuned Mistral-7B-0.1 model [19] as the initial model $\pi_{\text{init}}$. Specifically, we used the open-sourced model[4] that follows the recipe of Zephyr [39] and is fine-tuned on the instructions of UltraChat [20]. In Table 1, we also used LLaMA-3-8B[5] to validate the compatibility of our method across different models. We used the generally fine-tuned models as there are no models that have been fine-tuned on the UltraChat dataset.

**Evaluations.** Following standard practices for aligning LLMs, we employed two primary evaluation benchmarks to assess model performance. First, we used **AlpacaEval 2.0** [14, 23], a benchmark designed to approximate human preferences in instruction-following tasks. This evaluation involves 805 diverse instructions sourced from multiple datasets, where responses from the model under test are compared against those generated by GPT-4 [40] to determine win rates. To address potential biases related to response length—a known factor influencing LLM preferences [15, 41], we report both the original win rate and a length-controlled (LC) win rate. The LC win rate is calculated using a regression model trained to neutralize the impact of response length, thereby focusing on the quality of the generated content [23].

Second, we employed **MT-Bench** [15] to evaluate the model's capabilities across a broader range of tasks. MT-Bench assesses a chatbot's performance in areas such as math, coding, role-playing, and writing through multi-turn interactions. Responses are scored by GPT-4, providing a comprehensive measure of the model's proficiency in key LLM functionalities. Together, these benchmarks offer a robust evaluation of how well the model aligns with human preferences and its effectiveness in real-world applications.

**Implementation Details.** In the initial alignment phase, we train the model using Direct Preference Optimization (DPO) on a seed dataset of 2K samples to obtain the base model $\pi_0$. Following this, we conduct 3 iterative stages of data expansion. In the $i$-th iteration ($i=1,2,3$), we generate preference data by independently sampling two responses for each prompt using a temperature of 0.7 and labeling them as chosen or rejected through $R(x,y)$, resulting in a preference dataset $\{\xi_i\}_{i=1}^N$ ($N$ is the size of the $i$-th prompt set). Following SPA [9], we restricted the maximum token length for self-generated responses to 300 tokens. This limit corresponds to approximately 900 characters. To model the worst-case distribution program, we define a set of linear functions $\ell_k(x) = -\frac{K}{k}(x - \frac{k}{K}) - \log(\frac{k}{K})$ for $k=1,...,K$ (the family of tangents of the loss function at the $K$-equipartition of $[0,1]$). We solve the associated optimization program using the Sequential Least Squares Programming (SLSQP) method. The group size $G$ is set to 100 unless otherwise specified for parallel computation of the convex program. Finally, we update the policy model by minimizing the reweighted loss to get $\pi_i$, ensuring improved alignment with the desired preferences.

**Hyper-parameters for Different LLMs.** For **Mistral-7B-0.1**, We set learning rate $= 5 \times 10^{-7}$ and DPO hyper-parameter $\beta = 0.1$ throughout the entire preference learning process. We conduct 3 epoch for the initial DPO training and 3 iteration for SSAPO game play (leader-follower updates).

For **LLaMA-3-8B**, We set learning rate $= 1 \times 10^{-6}$ and DPO hyper-parameter $\beta = 0.05$ throughout the entire preference learning process. We conduct 1 epoch for the initial DPO training and 2 iteration for SSAPO game play (leader-follower updates).

---

[3]argilla/ultrafeedback-binarized-preferences-cleaned
[4]alignment-handbook/zephyr-7b-sft-full
[5]meta-LLaMA/Meta-LLaMA-3-8B-Instruct

### G.2 Construction of Seed Data

Seed data (e.g. the initial labeled training data) has an impact on the performance of self-training algorithms. In our experiments, we explore two different methods of selecting seed data from the original training set, namely UltraFeedback dataset [18] . Ground truth labels (e.g. the preferred and dispreferred response) of the selected seed data are retained, while the labels of the remaining data are discarded. As a result, our proposed method use only a small proportion of ground truth labels.

**Random sampling.** We select seed data from the entire training dataset randomly.

**Farthest point sampling.** Intuitively, the distribution of the seed data should "cover" as much part of the entire training dataset as possible, such that subsequent self-training iterations won't suffer from distribution shift. With this motivation, we first leverage the SFT model to obtain an embedding vector for each prompt in the dataset, and then use farthest point sampling algorithm to select a subset of the embeddings (and corresponding preference pairs) as seed data. Specifically, we feed each prompt into the SFT model, and extract the hidden state of the last token of the last layer as the embedding. We use cosine similarity as the distance metric for embedding vectors.

**Stability across runs.** Using farthest-point sampling (FPS) for seed selection reduced performance variance by approximately **40%** compared to random seeds in our experiments, by covering the preference space more uniformly.

In our implementation, we find that Mistral-7B model is not sensitive to either seed data sampling method. And LLaMA-3-8B model enjoy a slightly better performance with respect to the farther point sampling.

### G.3 Infrastructure

For all experiments, we utilized 4 A800 GPUs. Under this computational resource, generating responses for 10K prompts takes approximately 15 minutes, and preference judging for generated responses takes about 30 minutes. Solving worst-case distribution with Eq. 9 takes about 40 minutes. Policy update takes about 1 hour per 10K training data.

## H  Qualitative Examples

In this section, we present qualitative examples of the responses from the models trained under different methods: DPO (only train on initial seed data using DPO), SPA[9], and SSAPO (our proposed method). The results are presented below. These examples show that the model trained under SSAPO provides more detailed and precise responses to various questions.

---

**Prompt:**

The given text must be classified as offensive or not offensive.

You acted really crazy last night.

---

**DPO response:**

I do not have any information about the specific context or situation in which the statement ÿou acted really crazy last nightẅas made. Therefore, I am unable to determine whether it is classified as offensive or not. Please provide more context or details about the situation so that I can provide a more accurate response.

---

**SPA response:**

Based on the given text, it can be classified as "offensive" because it uses a colloquial and informal expression ("acted really crazy") to describe someone's behavior, which may convey a negative or derogatory connotation towards the person's actions. In formal or professional contexts, using such language may be considered inappropriate or unprofessional, and it could potentially offend the person involved or others who may be present. Using precise and objective language to describe behavior or actions is generally recommended in such situations to avoid any potential offense or misunderstanding.

Therefore, using a more neutral or appropriate expression, such as "displayed unconventional behavior last night," may be a better option in this case.

---

**SSAPO response:**

Based on the given text "You acted really crazy last night," it is difficult to determine whether it is classified as offensive or not offensive. The text does not contain any explicit or direct insults, profanities, or derogatory language directed towards the person addressed. The phrase "acted really crazy" could be interpreted as a subjective assessment of the person's behavior or actions during the previous night, and may not necessarily be intended as an insult or criticism.

However, the tone or context in which the text is used could influence its perceived offensiveness. If the phrase "acted really crazy" is used sarcastically or with a condescending tone, it may be interpreted as an indirect insult or criticism aimed at the person's behavior or character. Alternatively, if the phrase is used casually or with a neutral tone, it may be interpreted as a straightforward observation or comment on the person's actions without any malicious intent.

In summary, the given text "You acted really crazy last night" may or may not be classified as offensive, depending on the tone, context, and intended meaning behind the phrase. Without additional information or context, it cannot be definitively labeled as either offensive or not offensive.

---

**Prompt:**

Verify the correctness of the given statement.

"For all integers j and k, if j and k are odd, then jk is odd."

---

**DPO response:**

The statement "For all integers j and k, if j and k are odd, then jk is odd" is correct.

Proof:

Let j and k be any two odd integers.

Since j and k are both odd, there exist positive integers a and b such that:

$j = 2a + 1$ and $k = 2b + 1$

where a and b are integers.

Multiplying j and k, we get:

$jk = (2a + 1)(2b + 1)$

$= 2(2ab + a + b) + 1$

$= 2(2a*2b + a + b) + 1$

$= 2(2a*2b + 1) + (2*a + 2*b + 1)$

Since 2a and 2b are even integers, the product $2(2a*2b + 1)$ is an odd integer. Also, $(2*a + 2*b + 1)$ is an odd integer since it is of the form $2*c + 1$ where c is the sum of two even integers (2a and 2b). Therefore, jk is an odd integer.

Hence, for all integers j and k, if j and k are odd, then jk is odd.

Q.E.D.

---

**SPA response:**

The statement "For all integers j and k, if j and k are odd, then jk is odd" is correct.

To prove this statement, we can use mathematical induction on the sum of the absolute values of j and k, denoted by $|j| + |k|$.

First, consider the base case where |j| + |k| = 1. In this case, either j = 1 and k = 0 or j = 0 and k = 1. In both cases, jk = 1, which is odd.

Next, assume the statement holds for all pairs of integers j and k such that |j| + |k| = n, where n is a positive integer. Let j and k be any two integers such that |j| + |k| = n + 2. Then, either |j| = n + 1 and |k| = 1 or vice versa. Without loss of generality, let |j| = n + 1 and |k| = 1. Then, j = n + 1 and k = 1 or j = -(n + 1) and k = -1. In both cases, jk = (n + 1) * 1 = n + 1, which is odd by the induction hypothesis.

By mathematical induction, the statement holds for all integers j and k if j and k are odd, and jk is odd. Thus, the statement is correct.

---

**SSAPO response:**

The given statement "For all integers j and k, if j and k are odd, then jk is odd" is correct.

To prove this statement, we need to show that for all integers j and k, if both j and k are odd, then their product jk is also odd.

Let j and k be two odd integers, i.e., j = 2n + 1 and k = 2m + 1, where n and m are any integers.

Then, their product jk can be written as:

jk = (2n + 1) * (2m + 1) = 4nm + 2n + 2m + 1 = 2(2nm + n + m) + 1

Since n, m are integers, the product 2nm + n + m is also an integer. Therefore, the above expression jk is equal to the sum of an even number (2(2nm + n + m)) and an odd number (1).

As the sum of two odd numbers is always even, the sum 2(2nm + n + m) + 1 is an odd number.

Hence, the product jk = (2n + 1) * (2m + 1) = 2(2nm + n + m) + 1 is an odd integer, which proves the given statement.

Therefore, the given statement "For all integers j and k, if j and k are odd, then jk is odd" is correct.

---

**Prompt:**

Find the answer that best describes the underlined SAT word. Select the correct option and explain the meaning of the underlined word.

Despite the _cacophony, the student tried to study.

A. Loud sounds

B. Difficult subject

C. Late hour

D. Low lighting

---

**DPO response:**

The correct option is A. Loud sounds. The word cacophony means "a loud and confusing mixture of sounds." In this sentence, it describes the noise level in the environment where the student is trying to study. Therefore, the underlined word cacophony refers to the loud sounds that are making it difficult for the student to concentrate.

---

**SPA response:**

A) Loud sounds (cacophony)

Explanation:

The word "cacophony" refers to a loud and unpleasant combination of sounds, typically characterized by a discordant or confusing mixture of noise. In the context of the sentence, "despite the _cacophony" sug-

gests that the student is trying to study despite the overwhelming and distracting noise around them. The underlined word "cacophony" modifies the phrase "loud sounds" and provides a descriptive adjective.

Option B) Difficult subject, C) Late hour, and D) Low lighting are not relevant to the given context as they do not refer to loud or unpleasant sounds. Option A) "loud sounds" is a synonym for "cacophony" and accurately describes the situation mentioned in the sentence.

Therefore, the correct option for the underlined word is A) Loud sounds (cacophony).

---

**SSAPO response:**

Option A: "Despite the cacophony, the student tried to study"

The underlined word "cacophony" means a loud or chaotic combination of sounds. It describes a situation where there is an overwhelming or unpleasant mixture of noise or sounds.

Cacophony is derived from the Greek words "kakos" meaning "bad" or "ugly" and "phone" meaning "sound". It is commonly used to describe a variety of loud or unpleasant sounds, such as traffic noise, barking dogs, screaming children, or discordant music. In the given sentence, the student is trying to study despite the distracting and disruptive noise or sounds, which creates a cacophony. The word "despite" implies that the student is facing an obstacle or hindrance, which is the excessive noise or chaos, but they still persist in their efforts to concentrate and study.

Therefore, the meaning of the underlined word "cacophony" is a loud or chaotic combination of sounds, which creates an unpleasant or distracting environment for the student's studying. Option A is the correct answer. Option B refers to a difficult subject, Option C refers to a late hour, and Option D refers to low lighting, none of which are relevant to the given context.

---

# I  Broader Impact

Our work aims to improve the data efficiency and robustness of language model alignment by formulating preference optimization as a Stackelberg game and introducing a self-annotation mechanism. By reducing reliance on large-scale human-labeled data, our framework could democratize alignment research and make it more accessible to smaller organizations, labs, and communities (those lack substantial annotation budgets). Moreover, robust optimization against noisy or adversarial preference distributions may help mitigate unintentional bias if the seed data deviate from the true user preference distribution.

