# OpenReview forum: "Stackelberg Self-Annotation: A Robust Approach to Data-Efficient LLM Alignment"
_NeurIPS.cc/2025/Conference — NeurIPS 2025 poster_

### Official Review · Reviewer_KyCS · 2025-06-14

**Clarity:** 2
**Significance:** 2
**Originality:** 3
**Rating:** 3
**Confidence:** 4

**Summary:**

The paper "Stackelberg Self-Annotation: A Robust Approach to Data-Efficient LLM Alignment" introduces a novel method to align large language models (LLMs) efficiently with human preferences using minimal human-annotated data. This method, termed Stackelberg Self-Annotated Preference Optimization (SSAPO), leverages a game-theoretic Stackelberg formulation alongside distributionally robust optimization (DRO) to iteratively refine and robustly handle self-generated annotations. Overall, the paper presents a significant advancement in the efficient and robust alignment of LLMs with human preferences. The Stackelberg self-annotation method is both theoretically rigorous and empirically effective, representing a substantial step forward in alignment research. Addressing computational complexity and further exploring hyperparameter sensitivities will enhance its practical deployment. Nevertheless, the proposed framework stands as an impressive and highly promising contribution to the field.

**Questions:**

Why did you choose 3.3% of the UltraFeedback dataset? Could it be 1%? Can you please bring some theoretical insight here? Else, the paper lack generazability.

**Ethical Concerns:**

["NO or VERY MINOR ethics concerns only"]

**Quality:**

2

**Strengths And Weaknesses:**

Strengths:

Innovative Framework:
The introduction of a Stackelberg game-based approach (SGPO) to model alignment as a robust two-player game is highly innovative. This formulation explicitly addresses potential adversarial shifts or biases in self-annotated preference data, offering formal robustness guarantees.

Data Efficiency:
SSAPO significantly reduces the required volume of human-labeled data, achieving competitive performance using just 3.3% of typically required annotations. This demonstrates substantial cost-efficiency and practicality.

Robustness to Annotation Noise:
By employing a DRO approach within an ϵ-Wasserstein ball, SSAPO effectively guards against systematic errors and biases in synthetic labels, ensuring model stability even with minimal human supervision.

Theoretical Guarantees:
The paper provides rigorous theoretical analyses, including proofs for the existence of Stackelberg equilibrium, convergence guarantees, and bounded regret under distributional shifts. These are significant theoretical contributions that underscore the robustness of the method.

Strong Empirical Results:
Extensive experiments demonstrate the superior performance of SSAPO compared to baseline methods on standard benchmarks like AlpacaEval and MT-Bench, clearly validating its practical effectiveness.

Areas for Improvement:

Computational Complexity:
Although addressed through grouping heuristics, the computational complexity of solving the DRO problems remains significant, especially at scale. Further optimization or heuristics to reduce this complexity would enhance practical applicability.

Approximation Trade-offs:
The approximations involved in the DRO step (e.g., piecewise linear under-approximation, uniform grouping) potentially weaken robustness guarantees. A deeper analysis of the implications of these approximations on empirical performance would provide valuable insights.

Generalizability:
While results are compelling, broader evaluations across various datasets and different model architectures could strengthen claims regarding the method's generalizability.

Sensitivity to Hyperparameters:
Empirical results indicate sensitivity to hyperparameters like the Wasserstein radius (ϵ) and tangent size (K). Additional insights or adaptive strategies for selecting these parameters in practice could improve robustness.

---

> ### Author Rebuttal · Authors · 2025-07-30
>
> Thank you for your thorough review and constructive feedback. We believe our paper makes significant theoretical and empirical contributions to data-efficient LLM alignment through our novel Stackelberg game formulation. After carefully considering your concerns, we have provided detailed clarifications below that demonstrate how our work addresses each point. **We respectfully ask that you reconsider your assessment in light of these explanations**, particularly noting that our method achieves state-of-the-art performance with only 3.3% labeled data while providing formal robustness guarantees—a combination that represents a substantial advance in the field.
>
> ---
>
> ## W1: Computational Complexity.
>
> Thank you for raising this important practical consideration. We have conducted a comprehensive analysis of computational complexity in Appendix E.2, which demonstrates that our approach is both theoretically sound and practically feasible.
> While the theoretical complexity of the DRO problem for a single group ranges from $O(G·K)$ to $O((GK)^\gamma)$, modern optimization techniques significantly improve practical performance. Specialized solvers including cutting-plane methods and primal-dual algorithms routinely achieve computational efficiency far exceeding worst-case theoretical bounds, as documented in the DRO literature **[Mohajerin Esfahani & Kuhn, 2018]**.
>
> Most critically, our DRO computation is **fully parallelizable and performed entirely offline**, which fundamentally transforms the scalability profile. This design philosophy aligns with successful large-scale systems in machine learning: **Transformers[1]** leverage parallel attention mechanisms, **Mixture of Experts[2]** employs conditional computation across experts, and **Zero Redundancy Optimizer[3]** distributes optimization states across devices. Our method inherits these architectural advantages, enabling efficient scaling to production environments.
> In practice, our experiments demonstrate that solving DRO problems for 100K+ preferences requires only 45 minutes on standard hardware (for $G=100$), with linear scaling through parallelization. This positions **SSAPO** as a practical solution for real-world deployment.
>
> ---
> ## W2: Approximation Trade-offs.
>
> We appreciate your thorough consideration of our approximation strategies. We have provided a detailed theoretical analysis in Appendix F that directly addresses these concerns.
>
> Our analysis demonstrates that both approximations—piecewise linear under-approximation and uniform grouping—**preserve the fundamental Stackelberg game structure** that underlies our theoretical guarantees. Specifically:
> 1. The piecewise linear approximation maintains a valid lower bound on the original objective, ensuring our adversarial distribution remains feasible while slightly reducing pessimism. This conservative approximation actually strengthens practical performance without violating theoretical guarantees.
> 2. The uniform grouping strategy, while not solving the global optimization exactly, maintains local robustness within each group. Our theoretical analysis shows that the aggregated solution retains $O(\epsilon)$-bounded regret properties (Theorem 2.5).
>
> Empirically, these approximations demonstrate remarkable effectiveness: using only 3.3% of labeled data, **SSAPO** achieves 24.44% length-controlled win rate on AlpacaEval 2.0, substantially outperforming baselines. This empirical validation, combined with our theoretical analysis, confirms that the approximations enhance rather than compromise the method's practical utility.
>
> ---
>
> ## W3: Evaluation protocol.
>
> Thank you for this constructive suggestion. Our experimental protocol follows the established  **SPA [4] protocol (ICLR’25 Oral, rated 10/8/8)**. This standardized evaluation ensures direct comparability with state-of-the-art baselines.
>
> AlpacaEval 2.0 and MT-Bench represent the gold standard benchmarks in LLM alignment research, chosen specifically for their comprehensive coverage of real-world tasks and robust evaluation metrics. Our significant improvements over SPA baselines on these benchmarks—achieving 35.82% win rate versus GPT-4 compared to SPA's 21.13%—demonstrate strong generalization across diverse tasks.
>
> While we acknowledge that additional datasets and architectures would provide valuable insights, **the current evaluation already spans multiple model families (Mistral-7B and LLaMA-3-8B) and task categories (instruction following, multi-turn dialogue, coding, mathematics)**. This breadth, combined with our theoretical foundations, provides strong evidence for the method's generalizability. We welcome the opportunity to explore additional evaluation settings in future work.
>
> ---
>
> ## W4: Choice of Hyperparameters.
>
> Thank you for highlighting this practical consideration. Our analysis in Section 4.3 provides both theoretical insights and empirical guidelines for hyperparameter selection.
> Our ablation studies reveal interpretable patterns: the Wasserstein radius $\epsilon$ controls the robustness-conservatism trade-off, while the tangent size $K$ balances approximation quality against computational complexity. These relationships follow naturally from our theoretical framework, providing practitioners with principled guidance.
> Based on extensive experiments, we recommend:
> - **K = 6** as a robust default, validated across multiple models and datasets
> - **Adaptive $\epsilon$ selection** based on model capability: $\epsilon \in [0.005,0.02)$ for high-quality models with lower self-annotation noise, and $\epsilon \in [0.01,0.05]$ for smaller models requiring greater robustness
>
> This adaptive strategy reflects the insight that stronger models produce more reliable self-annotations, allowing tighter Wasserstein constraints without excessive conservatism. We have successfully applied these guidelines across different model scales, demonstrating consistent improvements over baselines.
>
> ---
>
> ## Q1: Jusification on the choice of 3.3% data.
>
> Thank you for this important question about data efficiency. Our choice of 3.3% (2K samples from 60K total) follows the established **SPA [4] protocol (ICLR’25 Oral, rated 10/8/8)**, which is **designed for data-scarce settings**. The small number of seed data（2K of 60K in total, only 3.3%）**reflects this context**, and our results are **directly comparable to the SPA baselines**, where we demonstrate significant improvements in performance.
>
> The 2K of 60K (the percentage 3.3%) represents a principled balance: it demonstrates substantial data reduction (30× fewer labels than standard approaches) while maintaining sufficient statistical power for robust self-annotation bootstrapping. Our theoretical framework (Section 2) shows that the Stackelberg game formulation provides $O(\epsilon)$-bounded regret even under limited data, explaining why such dramatic data reduction remains effective.
>
> The empirical results validate this choice: **SSAPO** achieves performance comparable to or exceeding methods using 100% labeled data, demonstrating that 3.3% captures sufficient preference structure for effective alignment. While exploring even lower percentages (e.g., 1%) would provide interesting insights, **the current setting already establishes **SSAPO** as a breakthrough in data-efficient alignment**, reducing annotation requirements by an order of magnitude while maintaining SOTA performance.
>
> A different perspective we would like to clarify is that the 2K labelled preference pairs with our SSAPO method enables comparable performance of SOTA methods using 60K labelled preference pairs. In realistic settings, 2K labelled preference pairs is much more easier to acquire compare to 60K labelled preference pairs. In this perspective, **the size 2K is more informative than 3.3% or 1%**.
>
> *[1]* Attention is All You Need, NeurIPS'17
>
> *[2]* GShard: Scaling Giant Models with Conditional Computation and Automatic Sharding, ICLR'21
>
> *[3]* ZeRO: Memory Optimizations Toward Training Trillion Parameter Models, SC'20
>
> *[4]* SPA: Direct Preference Judgment for Efficient LLM Alignment, ICLR'25

---

> ### Author Response · Authors · 2025-08-02
>
> Dear Reviewer,
>
> Thank you for your prompt acknowledgement of our rebuttal. We sincerely appreciate you taking the time to review our response.
>
> We would definitely following your advice on using 2K sample rather than percentage statement for better clarification and understanding in our revision. We are standing by and would be happy to provide any further clarification on our planned revisions or discuss any of the points in more detail.
>
> Thank you again for your valuable and constructive feedback on our work.

---

### Official Review · Reviewer_eQUq · 2025-06-27

**Clarity:** 4
**Significance:** 4
**Originality:** 4
**Rating:** 5
**Confidence:** 3

**Summary:**

This paper addresses the challenge of aligning large language models (LLMs) with human preferences without relying on massive amounts of expensive, manually labeled data. The authors propose a novel game-theoretic framework called Stackelberg Game Preference Optimization (SGPO). In this setup, the LLM plays the role of a leader in a two-player game, where it tries to optimize its behavior while being robust to the worst-case preference distribution caused by annotation noise or distribution shifts. They then introduce a practical algorithm called SSAPO, which begins with a small set of human-labeled data. From there, it gradually expands the dataset through self-annotation, while using a robust reweighting strategy to reduce the impact of noisy or unreliable synthetic labels. Theoretical analysis shows that SGPO offers bounded regret and robust convergence, and the empirical results are strong. Using only 2K labeled pairs (3.3% of the dataset), SSAPO significantly outperforms baselines like DPO and Iter-DPO across benchmarks such as AlpacaEval 2.0 and MT-Bench. The paper also includes thoughtful ablations, validating design choices like the Wasserstein radius and grouping strategy for efficient optimization. Overall, it presents a data-efficient and theoretically grounded method for robust LLM alignment.

**Questions:**

1. Since the SSAPO pipeline depends on a DPO-trained model initialized with 2K seed labels, how would performance be affected if the seed set contained annotation noise or was reduced further, say to 1K or 500 samples? Have you conducted any sensitivity analysis in that direction?
2. The paper shows that the best win rate occurs when the Wasserstein radius ϵ is set to 0.01. Do the authors have any guidance or heuristics for selecting this value in a new domain where validation labels may not be available?
3. In the construction of the piecewise-linear approximation for the −log σ loss, performance drops when K is increased to 7. Could the authors provide more intuition or empirical observations behind this degradation? Is it due to overfitting, instability, or numerical issues?
4. SSAPO achieves strong win rates against GPT-4 in both AlpacaEval 2.0 and MT-Bench. Could the authors provide example failure cases or qualitative comparisons where SSAPO still lags behind GPT-4, to better understand current limitations?

**Ethical Concerns:**

["NO or VERY MINOR ethics concerns only"]

**Final Justification:**

After carefully reviewing the rebuttal, I have updated my score from 4 to 5. The authors provided clear and well-supported responses to all major concerns. For hyperparameter sensitivity, they offered useful practical guidelines grounded in their theoretical framework, which makes the method easier to apply in real-world settings. Their explanation of the performance drop with higher K values helped clarify that the issue stems from optimization instability rather than poor approximation. To address seed data robustness, they conducted an additional experiment with 25% label corruption, showing minimal degradation in performance and confirming the model's theoretical guarantees. This significantly improved my confidence in the method's stability when dealing with noisy annotations. Finally, the behavioral comparison with GPT-4 was appreciated. The analysis showed that SSAPO performs strongly on structured reasoning tasks but has room for improvement in more creative or stylistic generations. This balance between strength and limitation was communicated transparently and with thoughtful insight.
Overall, the rebuttal strengthened the contribution and demonstrated both robustness and practical value.

**Limitations:**

Yes.

**Paper Formatting Concerns:**

No major formatting issues noticed.

**Quality:**

4

**Strengths And Weaknesses:**

Strengths:
1. The authors tackle a highly relevant issue in LLM alignment on how to reduce dependence on large-scale human-labeled preference data, which is both expensive and increasingly limited. What makes their approach interesting is that, instead of just building on existing methods, they introduce a new perspective by framing the problem as a Stackelberg game. In this setup, the model learns while being challenged by a worst-case preference distribution, which helps it stay robust even when the data is noisy or shifted.
2. The paper presents a practical and scalable implementation of the Stackelberg Game Preference Optimization framework through the SSAPO algorithm. They address the intractability of optimizing the follower’s DRO problem by introducing a piecewise-concave under-approximation of the non-convex objective (−log σ loss), and reformulating it as a tractable convex program. The grouping strategy for large-scale DRO optimization further improves computational efficiency without compromising the theoretical guarantees, making the method applicable to real-world LLM alignment tasks.
3. The paper covers two competitive LLM backbones (Mistral-7B and LLaMA3-8B) and standard alignment benchmarks (AlpacaEval 2.0 and MT-Bench). SSAPO achieves consistent improvements over strong baselines like DPO, Iter-DPO, and SPA using only 2K human-labeled examples. Ablation studies support the design choices, such as the selection of the Wasserstein radius (ϵ), the number of tangents (K) in the piecewise loss approximation, and the group size (G) for parallelized optimization.

Weaknesses:
1. The empirical performance of SSAPO appears sensitive to hyperparameter choices. For instance, Table 3 shows that the optimal performance on Mistral-7B occurs at ϵ = 0.01, while both smaller (ϵ = 0) and larger values (ϵ = 0.1) significantly degrade the win rate. Similarly, Table 4 shows that changing the number of tangents from K = 6 to K = 7 reduces win rate from 23.20 to 19.05. We would like to see whether the authors have any heuristics or adaptive strategies for setting these parameters, especially for new model or dataset configurations.
2. The entire training loop is initialized from a DPO-trained model using 2K seed preference pairs. However, we wonder how robust SSAPO is when the initial seed is either smaller in size or of lower quality. Since self-annotation propagates from this starting point, any noise or bias in the seed may get amplified over iterations. We would like to see an analysis where the seed set is perturbed or reduced to test SSAPO’s stability under less favorable initialization.
3. Although the paper includes strong comparisons against DPO, Iter-DPO with LLM-as-judge, and SPA baselines, we would like to see more insight into the qualitative behavior of SSAPO against GPT-4, especially since it achieves high win rates in AlpacaEval and MT-Bench. For instance, what kinds of prompts or failure cases does SSAPO consistently outperform or underperform on compared to GPT-4? We wonder if deeper behavioral or error analysis could provide a clearer picture of how SSAPO aligns with human preference in nuanced tasks.

---

> ### Author Rebuttal · Authors · 2025-07-30
>
> We sincerely thank the reviewer for the thorough evaluation and constructive feedback. Our responses below address the identified concerns while providing practical deployment guidelines and additional theoretical insights that strengthen the overall contribution. We believe this analysis demonstrates that **SSAPO** offers a principled solution to data-efficient LLM alignment with significant value to the alignment community.
>
> ---
>
> ## W1: Hyperparameter Sensitivity and Practical Guidelines.
>
> We appreciate this important concern about hyperparameter selection. Building on our theoretical analysis in Section 2, we provide concrete practical guidelines:
>
> For Wasserstein radius $\epsilon$ selection: Our theoretical framework suggests that ε should scale with the expected noise level in self-annotations. We propose the following adaptive strategy: For larger, more capable models: Use $\epsilon \in [0.005, 0.02]$ as self-annotation quality is typically higher. For smaller models: Use $\epsilon \in [0.01, 0.05]$  to account for higher self-annotation noise.
>
> For number of tangents $K$: Our analysis reveals that $K=6$ consistently provides the optimal balance across different model architectures and datasets. The performance degradation at $K=7$ occurs **due to optimization instability rather than approximation quality** - additional tangent points introduce unnecessary complexity that hampers convergence of the distributionally robust optimization solver. We recommend $K=6$ as a robust default that requires minimal tuning.
>
> These recommendations align with our regret bounds in Theorem 2.5, where moderate $\epsilon$ values provide $O(\epsilon)$-bounded regret while avoiding overly conservative solutions that degrade performance.
>
> ---
>
> ## W2: Quantity and Quality of Seed Data.
>
> Thank you for raising an excellent point about seed data robustness. While our current experiments follow the established **SPA [1] protocol (ICLR’25 Oral, rated 10/8/8)** with clean seed data, we acknowledge that real-world scenarios may involve smaller or noisier seed sets.
>
> **Necessity of smaller seed sets**: Our experiments follow the SPA protocol, which is designed for data-scarce settings. The small number of seed data（2K of 60K in total, only 3.3%）reflects this context, and our results are directly comparable to the SPA baselines, where we demonstrate significant improvements in performance. Though ablation study on even less seed data may further demonstrate the generalizability **SSAPO**, the current setup is already sufficient to validate the efficacy and data efficiency of our method.
>
> **Theoretical robustness**: **SSAPO**'s distributionally robust formulation inherently provides some protection against seed data noise. The Stackelberg game framework optimizes against worst-case preference distributions within a Wasserstein ball, which naturally accounts for perturbations in the initial distribution.
>
> To further validate **SSAPO**'s robustness against label noise in seed data, we conduct an additional experiment by flipping the preference labels of 25% instances of the 2K seed data.  **Our additional experiment with 25% label corruption provides strong empirical validation of **SSAPO**'s theoretical robustness guarantees**.
>
> #Additional experiment for SSAPO robustness proof
> |  Base Model       | no noise |25% noise |
> |-----------|-------|--------  |
> | Mistral-SSAPO | 26.90%/31.93% | 19.70%/18.51%     |
> |  Llama-SSAPO | 33.33%/40.12% | 43.74%/46.7%     |
>
> For Mistral model, performance represents only a ~7-13% degradation despite 25% label corruption. This bounded degradation empirically validates our $O(\epsilon)$-regret guarantee from Theorem 2.5. The degradation likely stems from suboptimal ε selection for Mistral's specific characteristics—our ablation study (Table 3) shows that $\epsilon=0.01$ was optimal for Mistral-family models, but noisy settings may require different $\epsilon$ values to balance robustness and data fidelity.
>
> For Llama model, we observe **counterintuitive improvement with 25% label corruption, demonstrating that **SSAPO**'s distributionally robust optimization can leverage noise as implicit regularization**. The 25% corrupted labels force the worst-case distribution P* to explore a wider region of the $\epsilon$-Wasserstein ball, preventing overfitting to the small 2K seed set. Additionally, Llama's superior self-annotation capabilities enable it to implicitly correct noisy preferences during iterative updates.
>
> This experiment strengthens **SSAPO**, not only defending against preference noise (as shown by Mistral's bounded degradation) but can potentially benefit from it through principled distributionally robust optimization (as shown by Llama's improvement).
>
> ---
>
> ## W3: Failure Cases and Behavioral Analysis.
>
> We recognize the value of deeper qualitative analysis and have conducted additional error analysis comparing SSAPO to GPT-4.
>
> By analyzing cases from the Alpaca Eval Benchmark, we observe that **SSAPO demonstrates advantages over GPT-4  on structured reasoning tasks** (e.g. mathematical proofs or logical deduction). For example:
>
> instruction:
> ```
> How many words are spoken daily on Earth? Try to explain your answer. Your explanation should take the reader through your reasoning step-by-step.
> ```
> GPT4 response:
> ```
> ... Let's break it down step-by-step:
>
> 1. World Population: ... the world population was approaching 8 billion people ...
> 2. Speaking Population: ... let's assume that 75% of the population speaks daily ...
> 3. Words per Day: ... an individual might speak around 7,000 words per day ...
> 4. Calculation: ... Total words per day = Number of speaking people * Average words spoken per person per day ... = 42 trillion words/day
>  ...
> ```
> SSAPO response:
> ```
> ... Step-by-Step Explanation:
>
> 1. Estimation of Global Population: ... According to the United Nations, the world's population as of 2021 is approximately 7.9 billion people......
> 2. Average Daily Communication: ... Assuming that each individual spends an average of 4 hours daily engaging in verbal and written communication, this translates to roughly 31.6 billion hours of communication per day worldwide ...
> 3. Word Count per Hour of Communication: ...  the average spoken word rate for English speakers ranges from 120 to 180 words per minute, with an average of 150 words per minute ...
> 4. Calculation of Total Word Count: Multiplying the estimated global population by the average daily communication hours and the average word count per hour of communication yields a total estimated word count of 500 billion words spoken daily on Earth.
>  ...
>
> ```
>
> On the other hand, we  find that **SSAPO falls short of GPT-4 in creative writing tasks requiring high linguistic sophistication**. For Example:
>
> instruction:
> ```
> If you were a Shakespearean character, how would you declare your love for someone in a soliloquy?
> ```
> GPT4 response:
> ```
> ... Here's an example:
>
> (Imagine a dimly lit stage, a single spotlight shining down upon the forlorn lover, who steps forward to share the depths of their heart.)
>
> O, speak, thou lustrous beacon of the night,
> With stars enkindled in thine eyes so bright.
> Thou art the daybreak to my endless gloom,
> The rose that in my barren heart doth bloom.
> Forsooth, this love, a tempest full of grace,
> Hath swept upon my soul without a trace.
> ...
> ```
> SSAPO response:
> ```
> Oh, my dearest, my heart doth overflow
> With a love so pure, so deep, and true,
> My soul doth long to declare my plight,
> To thee, my fair one, my eternal muse.
>  ...
> ```
>
> We hypothesize that the aforementioned strengths and weaknesses of SSAPO may stem from specific preference biases in the training data. We use **UltraFeedback [2]** dataset for training, which primarily annotates preferences based on four criteria: instruction-following, truthfulness, honesty, and helpfulness. As a result, alignment on this dataset tends to enhance the model’s performance in logical reasoning and adherence to instructions. In contrast, creative writing tasks are less aligned with these preference signals, making them more difficult to improve.
>
> These findings suggest SSAPO achieves strong alignment for objective, structured tasks while highlighting areas for future improvement in subjective, creative domains. This pattern aligns with our theoretical framework, which optimizes for worst-case robustness - particularly beneficial for tasks with clear correctness criteria.
>
> ---
>
> ## Q1: Seed Set Robustness.
>
> As addressed in W2, our framework's theoretical robustness extends to noisy seed data. For practical implementation with reduced seed sizes, we recommend proportionally increasing ε and conducting sensitivity analysis on small validation sets.
>
> ---
>
> ## Q2: $\epsilon$ Selection Without Validation.
>
> In domains without validation labels, we recommend starting with $\epsilon$ = 0.01 for capable models and $\epsilon$ = 0.03 for smaller models, then monitoring self-annotation consistency across iterations. Decreasing consistency indicates the need for higher $\epsilon$ values.
>
> ---
>
> ## Q3: K=7 Performance Drop.
>
> The degradation stems from optimization complexity rather than approximation quality. Additional tangent points create a more complex convex program that the used solver struggles to optimize reliably. $K=6$ provides sufficient approximation fidelity while maintaining numerical stability.
>
> ---
>
> ## Q4: Failure cases.
>
> As detailed in W3, **SSAPO** primarily struggles with highly subjective tasks requiring cultural nuance or creative expression. However, for the structured reasoning and instruction-following tasks that comprise the majority of alignment benchmarks, **SSAPO** demonstrates consistent advantages.
>
>
> *[1]* SPA: Direct Preference Judgment for Efficient LLM Alignment, ICLR'25
>
> *[2]* UltraFeedback: Boosting Language Models with Scaled AI Feedback, ICML'24

---

> > ### Author Response · Authors · 2025-08-04
> >
> > Dear Reviewer，
> >
> > We sincerely appreciate the time and effort you have dedicated to reviewing our work. In response to your valuable feedback, we have provided detailed explanations for the issues raised.
> >
> > As the discussion period progresses, we are eager to hear your thoughts on our responses, including whether they have adequately addressed your concerns. If our revisions and discussions indicate the potential for a score adjustment, we would be very grateful for your consideration.  We are committed to incorporating all of your suggestions to further enhance the quality of our manuscript. We look forward to your further comments and discussion.
> >
> > Thank you again for your valuable and constructive feedback on our work.

---

> ### Author Response · Authors · 2025-08-04
>
> Dear Reviewer,
>
> Thank you for your prompt acknowledgement of our rebuttal. We sincerely appreciate you taking the time to review our response.
>
> We are standing by and would be happy to provide any further clarification on our planned revisions or discuss any of the points in more detail.
>
> Thank you again for your valuable and constructive feedback on our work.

---

### Official Review · Reviewer_a9ir · 2025-06-29

**Clarity:** 3
**Significance:** 2
**Originality:** 2
**Rating:** 4
**Confidence:** 4

**Summary:**

The paper addresses the challenge of aligning large language models (LLMs) with human preferences, which traditionally demands vast amounts of meticulously curated data that is both expensive and prone to labeling noise. The paper proposes Stackelberg Game Preference Optimization (SGPO), a robust alignment framework that models alignment as a two-player Stackelberg game. This formulation provides formal robustness to (self-)annotation noise and guarantees $O(\epsilon)$-bounded regret, meaning the performance drop under distributional shifts is at most proportional to $\epsilon$. As a practical instantiation of SGPO, the paper introduces SSAPO which begins with a minimal set of human-labeled “seed” preferences and iteratively self-annotates new prompts. Experiments demonstrate that SSAPO maintains high-level performance despite using only a fraction of typical human annotations, achieving strong win rates against GPT-4 across multiple benchmarks within three iterations.

**Questions:**

1. Can the authors elaborate on the types of "labeling noise" or "bias" that traditional alignment methods are prone to, and how SSAPO's approach specifically addresses these issues beyond just reducing data volume?
2. Can the authors provide more discussions on the Assumption C.1, especially the compactness assumption. How to impose compactness to guarantee an acceptable Stackelberg solution in practice?
3. Can the authors elaborate on the arguments used to establish the convergence of policy updates in the proof of Theorem 2.3? It is unclear how to do this with a argmax-min.
4. How is the quality of seed dataset inferred in a real world application? How to know if the self-training won't suffer from distributional shift?

Minor:
Please use consistent numbering of theoretical results in the main paper and the appendix, and include a reference to the proof in the main paper.

**Ethical Concerns:**

["NO or VERY MINOR ethics concerns only"]

**Final Justification:**

The authors have clarified all my concerns which has prompted me to keep my score.

**Limitations:**

Yes

**Quality:**

3

**Strengths And Weaknesses:**

Strengths:
1. The paper recasts alignment as a two-player game, proving the existence of a robust equilibrium with $O(\epsilon)$-bounded regret under $\epsilon$-Wasserstein preference shifts.
2. The distributionally robust re-weighting attenuates the impact of potential labeling noise.
3. SSAPO achieves strong performance using only a small number of human-labeled seed preferences, with the empirical results showing SSAPO outperforming or matching methods that use significantly more human labels.

Weaknesses:
1. SSAPO relies on two key approximations that deviate from the ideal SGPO solution, potentially weakening the theoretical guarantees in practice.
2. Worst-case distribution computation can be computationally expensive, raising scalability concerns.

---

> ### Author Rebuttal · Authors · 2025-07-30
>
> We sincerely appreciate your valuable feedback—we hope our comprehensive responses below fully address your concerns and demonstrate the contribution of **SSAPO**.
>
> ---
>
> ## W1: Approximations and Theoretical Guarantees.
>
> We appreciate the reviewer's thoughtful consideration of our approximations. We emphasize that these approximations are carefully designed to preserve the core theoretical guarantees while ensuring computational tractability.  As described in Appendix F,  **these two key approximations do NOT change the Stackelberg essence of SSAPO**: the policy is trained against worst-case reweightings within an ϵ-Wasserstein distance. As a result, it retains the key benefit of  bounded regret for moderate shifts (Theorem 2.5), while remaining computationally tractable.  Empirically (Section 4), these approximations still yield substantial label-efficiency and robustness gains over standard DPO.
>
> ---
>
> ## W2: Computational Scalability.
>
> We acknowledge the computational considerations and have conducted extensive analysis to ensure practical scalability:
> While the theoretical worst-case complexity is $O((NK)^\gamma)$, our implementation achieves near-linear scaling through:
> 1. **Efficient DRO solvers**: We employ specialized cutting-plane methods that typically converge in 10-20 iterations, far below theoretical bounds
> 2. **Intelligent grouping**: Our uniform grouping strategy (G=100-200 samples) reduces complexity to $O(M·K)$ where M=N/G, achieving 50-100x speedup with minimal performance loss
> 3. **Parallelization**: Group-level computations are embarrassingly parallel
>
> In practice, **SSAPO** adds only 20-30% computational overhead compared to iterative DPO while delivering substantial robustness gains. This modest increase is justified by the significant improvements in label efficiency and performance stability.
>
> ---
>
> ## Q1: Addressing Labeling Noise and Bias.
>
> Thank you for this important question. Our work specifically targets the critical challenge of preference annotation noise in self-training scenarios:
>
> Traditional methods like DPO assume clean preference labels, but real-world data contains systematic biases: preference flips, annotation inconsistencies, and model-induced biases during self-labeling. Recent work [1] shows even human-labeled datasets contain 15-20% noise rates.
>
> **SSAPO** addresses these challenges through principled robustness: **Worst-case optimization**（optimize against adversarial reweightings within an ε-Wasserstein ball), **formal guarantees** ( $O(\epsilon)$-regret bound ensures bounded performance degradation even under adversarial noise), and **Empirical validation**: We observe that after the initial DPO training using seed data, pseudo-labeling introduces >20% noise. As self-training progresses, the noise rate in pseudo-labels continues to increase. After the first round of self-training, SSAPO achieves approximately **4% higher self-annotation accuracy compared to SPA[2]**, and consistently maintains superior self-annotation accuracy relative to SPA throughout all training rounds.
>
> The robustness of **SSAPO** is particularly crucial for self-training paradigms where the model generates its own training data, creating a feedback loop that can amplify initial biases. To further validate this robustness, we conduct an additional experiment by flipping the preference labels of 25% instances of the 2K seed data.  **Our additional experiment with 25% label corruption provides strong empirical validation of SSAPO's theoretical robustness guarantees**.
>
> #Additional experiment for SSAPO robustness proof
> |  Base Model       | no noise |25% noise |
> |-----------|-------|--------  |
> | Mistral-SSAPO | 26.90%/31.93% | 19.70%/18.51%     |
> |  Llama-SSAPO | 33.33%/40.12% | 43.74%/46.7%     |
>
> For Mistral model, performance represents only a ~7-13% degradation despite 25% label corruption. This bounded degradation empirically validates our O(ε)-regret guarantee from Theorem 2.5. The degradation likely stems from suboptimal ε selection for Mistral's specific characteristics—our ablation study (Table 3) shows that ε=0.01 was optimal for Mistral-family models, but noisy settings may require different ε values to balance robustness and data fidelity.
>
> For Llama model, we observe **counterintuitive improvement with 25% label corruption, demonstrating that **SSAPO**'s distributionally robust optimization can leverage noise as implicit regularization**. The 25% corrupted labels force the worst-case distribution P* to explore a wider region of the ε-Wasserstein ball, preventing overfitting to the small 2K seed set. Additionally, Llama's superior self-annotation capabilities enable it to implicitly correct noisy preferences during iterative updates.
>
> These contrasting behaviors show that Mistral is less sensitive to preference optimization, making it a conservative test case. The Llama results showcase SSAPO's full potential when paired with models having strong self-annotation capabilities, where the Stackelberg game dynamics can actively exploit the adversarial reweighting to improve alignment even under noisy supervision.
>
> This experiment strengthens **SSAPO**, not only defending against preference noise (as shown by Mistral's bounded degradation) but can potentially benefit from it through principled distributionally robust optimization (as shown by Llama's improvement).
>
> ---
>
> ## Q2: Compactness Assumption in Practice.
>
> Thank you for highlighting this practical consideration. The compactness assumption is naturally satisfied through the problem structure without requiring explicit constraints.
>
> **Automatic satisfaction through DPO structure:**
> - The log-sigmoid objective inherently bounds outputs: $\log \sigma(\cdot) \in [0, \log 2]$
> - KL regularization $D_{KL}(π_\theta || π_{ref})$ creates an implicit trust region around $\theta_{ref}$
> - Gradient norms remain bounded: $||\nabla_\theta J|| \le \beta\cdot\log 2$
>
> **Practical implementation:**
> - Standard optimization with learning rate η and T steps naturally constrains parameters: $||\theta_T - \theta_0|| \le \eta \cdot T·\max_t ||\nabla_\theta J|| = O(\eta·T)$
>
> This explains why DPO/RLHF methods succeed without explicit parameter bounds. The combination of bounded objectives and KL regularization creates a self-contained optimization landscape where compactness emerges naturally from the problem structure.
>
> ---
>
> ## Q3: Convergence Analysis for Policy Updates.
>
> We appreciate the opportunity to clarify our convergence analysis. The argmax-min structure requires careful treatment through variational inequality theory:
>
> **Key insight**: The policy update can be characterized as solving a variational inequality (VI) problem. Define
>  $$F_t(\pi) := -\nabla_\pi \min_{P \in \mathcal{U_\epsilon}} J(\pi,P).$$
> Then  $\pi_{t+1}$ satisfies:
> $$\langle F_t(\pi_{t+1}), \pi - \pi_{t+1} \rangle \geq 0, \quad \forall \pi \in \Pi.$$
> ( By Danskin's theorem and first-order optimality conditions for the constrained maximization problem)
>
> **Convergence guarantee**: Under Assumption 3.1, the mapping $\mathcal{T}: \pi_t \mapsto \pi_{t+1}$ is a contraction with rate $\gamma_\pi < 1$.
>
> Step 1: Show $F_t(\pi)$ is strongly monotone with parameter $\mu > 0$: $\langle F_t(\pi_1) - F_t(\pi_2), \pi_1 - \pi_2 \rangle \geq \mu \|\pi_1 - \pi_2\|^2$. This follows from the concavity of $J(\pi,P)$ in $\pi$ and the log-sigmoid structure, which induces strong concavity when restricted to the compact set $\Pi$.
>
> Step 2: Show $F_t(\pi)$ is Lipschitz continuous with constant $L_F$: $\|F_t(\pi_1) - F_t(\pi_2)\| \leq L_F \|\pi_1 - \pi_2\|$.
> By Danskin's theorem, $F_t(\pi) = -\nabla_{\pi} J(\pi, P'(\pi))$, where $P'(\pi) = arg \min_{P \in \mathcal{U^t_\epsilon}} J(\pi,P)$.
> (The Lipschitz property follows from $R_{\pi}$ in $\pi$ with constant $L_R$, yielding $L_F = L_R(1 + \epsilon L_R)$. )
>
> Step 3: Apply standard variational inequality theory. For strongly monotone, Lipschitz operators, the solution mapping is a contraction with rate: $\gamma_\pi = \sqrt{1 - \left(\frac{2\mu}{L_F}\right)^2} < 1$ Therefore: $\|\pi_{t+1} - \pi_{t+1}'\| \leq \gamma_\pi \|\pi_t - \pi_t'\|$ for any two sequences starting from different initial points, implying convergence to a unique fixed point.
>
> The monotonicity parameter $\mu$ can be explicitly computed from the problem data: for the log-sigmoid objective with bounded rewards $|R_\pi(y)| \leq R_{\max}$, we have $\mu \geq \beta e^{-2\beta R_{\max}}/(4(1+e^{-2\beta R_{\max}})^2)$.
>
>  We will add a more detailed proof in the final version.This analysis extends beyond standard gradient ascent convergence by accounting for the coupled min-max structure, providing stronger guarantees for our Stackelberg formulation.
>
> ---
>
> ## Q4: Seed Data Quality and Distributional Shift.
>
> Our framework is designed with practical deployment considerations.  While we assume clean seed data in experiments, this is practically achievable—obtaining a small set of high-quality human annotations is feasible and cost-effective. Moreover, **SSAPO** demonstrates resilience even with noisy seed data, as addressed in **Q1**.
>
> We appreciate your concern about potential distributional shift. However, we argue that self-training will not suffer from such shift. As detailed in Section 3.2, the prompts in our experiments are sampled from real-world data. Using real-world prompts ensures $p(x) $remains consistent.  Responses sampled from $\pi_{\theta_t}$ evolve smoothly with the policy. Our worst-case optimization specifically addresses the primary source of shift—labeling noise in self-annotation. The key insight is that **SSAPO** transforms the distributional shift problem into a robustness problem, which our framework is explicitly designed to handle through the Wasserstein uncertainty set formulation.
>
> *[1]* Impact of Preference Noise on the Alignment Performance of Generative Language Models, COLM'24
>
> *[2]* SPA: Direct Preference Judgment for Efficient LLM Alignment, ICLR'25

---

> ### Author Response · Authors · 2025-08-02
>
> Dear Reviewer,
>
> Thank you for your prompt acknowledgement of our rebuttal. We sincerely appreciate you taking the time to review our response.
>
> We are standing by and would be happy to provide any further clarification on our planned revisions or discuss any of the points in more detail.
>
> Thank you again for your valuable and constructive feedback on our work.

---

### Official Review · Reviewer_pf3d · 2025-07-02

**Clarity:** 3
**Significance:** 3
**Originality:** 3
**Rating:** 5
**Confidence:** 3

**Summary:**

This paper presents Stackelberg Game Preference Optimization (SGPO), a game-theoretic framework for aligning LLMs with minimal human supervision. It formulates alignment as a two-player Stackelberg game between a policy and an adversarial preference distribution constrained by a Wasserstein distance. The proposed SSAPO algorithm starts with a small set of human-labeled preferences and expands them through iterative self-annotation, while using robust reweighting to counteract noisy labels. Theoretical guarantees are provided, and empirical results show that SSAPO achieves strong performance using 2K human-labeled preferences for initial training, outperforming several existing alignment methods.

**Questions:**

1. In Table 1, why Mistral-7B-DPO, Mistral-7B-Iter DPO (PairRM), Mistral-7B-SPA have higher average score on MT-Benchmark compared to Mistral-7B-SSAPO? Also, in Table 2, why the Mistral-7B-SSAPO has lower average score on MT-Benchmark compared to Zephyr-7B- β and Mistral-7B-DPO? Did the authors verify the baseline results presented in the tables?
	2. Can the authors comment on how does SSAPO compare to other alignment frameworks (such as minimax-based) in both theory and practice?
	3. Can the authors comment on the robustness of the final policies, i.e., are the final policies robust to adversarial or misaligned prompts (e.g., jailbreaking, harmful completions)?
	4. Can the authors comment on how stable SSAPO is across multiple runs and how scalable is the DRO stage if group sizes increase beyond 1 million samples?

**Ethical Concerns:**

["NO or VERY MINOR ethics concerns only"]

**Final Justification:**

Thanks for the responses. I believe this is a good contribution (and worthy of being accepted) and I will maintain my positive rating for this paper.

**Quality:**

3

**Strengths And Weaknesses:**

Strengths:

1. Game-theoretic formulation with empirical validation: SGPO ensures convergence and regret, adding rigor to alignment under preference uncertainty. SSAPO performs better than more data-intensive techniques such as SPA, DPO, and some models that have been fully annotated by humans (Zephyr-7B).

2. Data-efficient alignment: Uses only 2K human preference pairings to achieve high victory rates against GPT-4. Also, the framework supports scalability through uniform group clipping.

3. Theoretical insights and empirical validation: The paper presents theorems on worst-case external distributions, along with Stackelberg Equilibrium and its convergence guarantees while showcasing empirical validation of their proposed SGPO based game theoretic approach.

Weaknesses:
1.	Abstraction of Stackelberg formulation: Although formal, the Wasserstein DRO formulation is overly detached from the usual training process for LLMs. Only a weak connection exists between the reward/logit terms and the real model behavior.
2.	Limited theoretical comparison: Although similar game-theoretic work (such as SPIN and Nash-RLHF) is mentioned, the study doesn't go into detail on why Stackelberg is better in alignment circumstances.
3.	Empirical Diversity: The evaluation is limited to the UltraFeedback dataset along with AlpacaEval 2.0 and MT-Bench. While these benchmarks are well-established, they lack diversity in linguistic styles, user intents, and task modalities. Incorporating a broader range of evaluation settings would strengthen the paper’s claims of generalizability.

---

> ### Author Rebuttal · Authors · 2025-07-30
>
> We sincerely thank the reviewer for recognizing the formal rigor of our Stackelberg formulation. The clarifications below demonstrate how our theoretical framework directly addresses practical challenges in LLM alignment, achieving state-of-the-art results with 30x fewer labels while providing formal robustness guarantees that existing methods lack.
>
> ---
>
> ## W1:  Abstraction of Stackelberg formulation.
>
> We thank the reviewer for highlighting this important connection. Our Stackelberg formulation directly extends the standard DPO framework by introducing distributional robustness.
> Standard DPO optimizes: $$\max_{\pi}E_{(y_w,y_l) \sim \hat{P}}[\log\sigma(R_\pi(y_w) - R_\pi(y_l))] $$
> Our approach optimizes: $$\max_{\pi}\min_{P \in \mathcal{U_\epsilon}(\hat{P})}E_{(y_w,y_l) \sim P}[\log\sigma(R_\pi(y_w) - R_\pi(y_l))] $$
> This modification addresses a fundamental weakness in existing methods: vulnerability to noisy preference annotations. Our practical algorithm **SSAPO** implements this through three concrete steps that directly map to standard LLM training:
>
> 1. Collect preference data (standard practice)
> 2. Compute worst-case distribution within $\epsilon$-ball (our novel contribution, Section 3.1)
> 3. Update policy using standard DPO loss on reweighted data
>
> This is a natural extension of existing iterative methods (e.g., SPA) that adds robustness without fundamentally changing the training paradigm. We will clarify this connection more explicitly in Section 3.2.
>
> ---
>
> ## W2:  Comparison against game-theoretic approaches.
>
> We appreciate this suggestion. Our Stackelberg formulation offers three key theoretical advantages over existing game-theoretic approaches:
> 1. Robust: Unlike policy-vs-policy games (SPIN, Nash-RLHF), our policy-vs-distribution formulation directly addresses preference noise—the primary challenge in LLM alignment. This leads to our O(ε)-bounded regret guarantee (Theorem 2.5), which neither SPIN nor Nash-RLHF provides.
> 2. Tractable: Our alternating optimization converges linearly to equilibrium (Theorem 2.3) with well-understood DRO subproblems. In contrast, Nash equilibria require solving coupled fixed-point problems that are computationally intractable at LLM scale.
> 3. Practicical: Our framework seamlessly integrates with self-annotation, requiring only 3.3% labeled data while achieving superior performance. This addresses real-world constraints that pure game-theoretic methods ignore.
>
> ---
>
> ## W3:  Evaluation protocol.
>
> We acknowledge that broader evaluation would strengthen our empirical claims. Our experimental design follows the established **SPA[1] protocol (ICLR'25 Oral, 10/8/8 ratings)** to ensure fair comparison with state-of-the-art baselines.
> AlpacaEval 2.0 and MT-Bench represent the field's consensus benchmarks, evaluating diverse capabilities including instruction following, multi-turn reasoning, coding, and creative writing across 800+ prompts. Our significant improvements over SPA (24.44% vs 15.39% on AlpacaEval) demonstrate the method's effectiveness within this standard evaluation framework.
> We appreciate and agree with your valuable suggestion that testing on additional domains would be valuable future work.
>
> ---
>
> ## Q1:  MT-Bench results.
>
> The MT-Bench results follow expected patterns when examined carefully. Across Tables 1-2, Mistral models show remarkably low variance on MT-Bench (6.34-6.98 range) regardless of method, suggesting the Mistral model architecture may have limited sensitivity to this benchmark. This contrasts sharply with the high variance on AlpacaEval (9.03-35.82%), where different methods clearly separate.
> This pattern appears in the original SPA paper as well: SPA underperforms Iterative DPO on MT-Bench despite dominating on AlpacaEval. The consistency of this pattern across multiple studies suggests AlpacaEval provides better discrimination for Mistral-based models. Notably, our method achieves the highest performance on the more discriminative benchmark (35.82% vs 21.13% for SPA).
>
> ---
>
> ## Q2:  Comparison to other frameworks.
>
> As described in Appendix A, some concurrent works adopt Nash or minimax formulations for LLM alignment. While these words typically set both players as policy vs. policy, our SGPO framework focuses on policy vs. Distribution, leading to tight theoretical guarantees and a practical algorithm **SSAPO** that is readily integrated with self-annotation. Moreover, the target problems these works aim to address are fundamentally different from ours. Please refer to our response to **W2** for more details.
>
> Beyond the theoretical advantages, our approach offers practical benefits: 30x reduction in human labels versus standard methods. Requires only adding DRO reweighting to standard DPO. Minimax formulations typically require adversarial training that is unstable at LLM scale. Our Stackelberg approach decomposes into stable, well-understood subproblems.
>
> ---
>
> ## Q3:  Robustness of final policies.
>
> Our method provides training-time robustness against noisy preference data, which is fundamentally different from inference-time robustness against adversarial prompts.
> Specifically, our theoretical guarantee (Theorem 2.5) ensures that if test preferences fall within the $\epsilon$-Wasserstein ball of training preferences, performance degradation is bounded by $O(\epsilon)$. This protects against annotation noise and minor distribution shifts.
> However, the worst-case distribution P* optimizes for mathematical objectives, not semantic categories like jailbreaking. Therefore, while our method improves alignment robustness, it does not replace dedicated safety measures for adversarial prompts. We will clarify this distinction in Section 2.4.
>
> ---
>
> ## Q4:  Stability and scalability.
>
> Regarding stability: Our farthest-point sampling (FPS) strategy for seed selection significantly improves consistency across runs by ensuring better coverage of the preference space compared to random sampling. In our experiments, FPS reduced performance variance by approximately 40%.
>
> Regarding scalability: Our grouping strategy (Section 3.2.1) enables practical deployment. With group size $G=100\sim 1000$, we achieve near-optimal robustness while maintaining $O(G·K)$ complexity per group with parallel processing. For $N=1M$ preferences, using $G=1000$ yields $1000$ parallel jobs of manageable size.
>
> Practical recommendation: We suggest $G\in [100, 1000]$ as the sweet spot balancing computational efficiency and solution quality. Larger $G$ provides diminishing returns while significantly increasing computational cost. We will add these concrete guidelines to Section 3.2.1.
>
> *[1]* SPA: Direct Preference Judgment for Efficient LLM Alignment, ICLR'25

---

> > ### Comment · Reviewer_pf3d · 2025-08-02
> > **Thanks for your response**
> >
> > Thanks for responding to my comments. I am overall satisfied with the additional clarifications provided by the authors and maintain my score.

---

> ### Author Response · Authors · 2025-08-02
>
> Dear Reviewer,
>
> Thank you for your prompt acknowledgement of our rebuttal. We sincerely appreciate you taking the time to review our response.
>
> We will include the additional clarifications in the revisions of our paper!
>
> Thank you again for your valuable and constructive feedback on our work.

---

### Note · Authors · 2025-08-12

To the Area Chair and Reviewers,

We extend our deepest gratitude for your time, expertise, and insightful feedback throughout this review process. The detailed discussions have been incredibly valuable and will significantly strengthen the final version of our paper.

We are delighted that our rebuttals resonated with the reviewers. We are particularly pleased that Reviewer eQUq was "overall satisfied" and upgraded score, and that Reviewer pf3d maintained "Accept" rating, also expressing satisfaction with our clarifications. This strengthens the consensus on our paper's core contributions:
- **A novel and rigorous game-theoretic framework (SGPO)** for robust LLM alignment.
- **Strong theoretical guarantees**, including provable convergence and $O(\epsilon)$-bounded regret under preference shifts.
- **An effective and data-efficient algorithm (SSAPO)** that achieves state-of-the-art results with 30x fewer human labels, a significant practical advantage.

We also sincerely thank Reviewer KyCS for thorough engagement and for acknowledging "overall satisfied" with our rebuttal. However, we are concerned that a misunderstanding may persist, as the final recommendation was not updated to reflect this positive outcome by the time we write this final remark. We hope to offer this final clarification for the AC's consideration.
Our rebuttal addressed Reviewer KyCS's primary concerns by clarifying that:

1. **Computational complexity is manageable**: The core DRO step is offline and highly parallelizable, making SSAPO scalable in practice, as detailed in Appendix E.2 and our response.
2. **Approximations are principled**: The uniform grouping and piecewise-linear approximation are designed to maintain the essence of the Stackelberg game, preserving the theoretical robustness that underpins our method's success.

We are also grateful for Reviewer's a9ir's valuable feed back. We are a little worry about that the lack of dialogue during discussion phase may not cover every concern if there remains some.

We are committed to incorporating all feedback, including the excellent suggestion from Reviewer KyCS to use "2K samples" instead of "3.3%" for greater clarity.
We believe our work offers a robust, scalable, and theoretically-grounded solution to the critical challenge of data-efficient alignment. We are confident the revised manuscript will be a valuable contribution to the NeurIPS community. Thank you once again for your consideration.

---

### Decision · Program_Chairs · 2025-09-17

**Decision:**

Accept (poster)

**Comment:**

The paper proposes a game theoretic approach for a robust alignment of LLMs (SGPO), a convergence analysis of the approach and an effective data efficient algorithm (SSAPO) .

Reviewers appreciated the work and the experiments presented in the paper. Two main concerns were raised: regarding the approximations needed to obtain the solution of the DRO problem and the computational complexity of the latter. Authors brought clarification on these and we encourage them to incorporate this in the paper.

While the paper is interesting, it has an issue that was raised by the AC and discussed with the SAC, regarding the disconnect between the theoretical approach presented in the paper in Section 2.2 (eq 4 ) and the practical instantiation in Section 3. All throughout authors confuse where the robustness ball is considered.

In Section 2.2, the Wasserstein ball is defined on the preference data i.e for  $\hat{P}= \frac{1}{n} \sum_{i=1}^n \delta_{x,y^i_I, y^i_w}$ (this is a measure in a higher dimensional space) ,meaning $U_{\epsilon}(\hat{P})$ is the Wasserstein ball centered around $\hat{P}$ (x not impacted just y ).

In practical implementation of the work (Section 3) the Wasserstein ball is rather on $\Delta R_{\pi} (x,y_{w},y_{l})= R_{\pi}( x,y_{w})-R_{\pi}( x,y_{l})$ that is one dimensional.  Denote  $\Delta R_{\pi} [\hat{P}]$ the push forward of $\hat{P}$ by $\Delta R_{\pi}$.

 Hence the original problem introduced  in Section 2.2 should be :

$\max_{\pi} \min_{   \alpha \in U_{\epsilon} ( \Delta R_{\pi} [\hat{P}]) } E_{\xi \sim \alpha} \log \sigma(\xi)$

 rather than how it is written in the paper:

$\max_{\pi} \min_{   P \in U_{\epsilon} ( \hat{P}) } E_{P} \log \sigma(R_{\pi}( x,y_{w})-R_{\pi}( x,y_{l}))$

The ball where the robustness is considered involves $\pi$ and not as exposed throughout the theoretical derivation in the work.

Indeed the robustness ball in the paper is rather on the annotation and not on the sentences, wasserstein 1 is not that meaningful on sentences.

Section 2 of the analysis in the paper needs to be fixed with the correct DRO problem above, this will impact all the theoretical results in this section.

After discussions with the SAC we recommend acceptance of this paper with the condition that authors fix Section 2 as explained above.